# ABL1-dependent OTULIN phosphorylation promotes genotoxic Wnt/β-catenin activation to enhance drug resistance in breast cancers

Wei Wang[1,2,3], Mingqi Li[2,3], Suriyan Ponnusamy[3,4], Yayun Chi[5,6], Jingyan Xue[5,6], Beshoy Fahmy[3], Meiyun Fan[2,3], Gustavo A. Miranda-Carboni[3,4], Ramesh Narayanan [3,4], Jiong Wu[5,6] & Zhao-Hui Wu [1,2,3 ✉]

Dysregulated Wnt/β-catenin activation plays a critical role in cancer progression, metastasis, and drug resistance. Genotoxic agents such as radiation and chemotherapeutics have been shown to activate the Wnt/β-catenin signaling although the underlying mechanism remains incompletely understood. Here, we show that genotoxic agent-activated Wnt/β-catenin signaling is independent of the FZD/LRP heterodimeric receptors and Wnt ligands. OTULIN, a linear linkage-specific deubiquitinase, is essential for the DNA damage-induced β-catenin activation. OTULIN inhibits linear ubiquitination of β-catenin, which attenuates its Lys48-linked ubiquitination and proteasomal degradation upon DNA damage. The association with β-catenin is enhanced by OTULIN Tyr56 phosphorylation, which depends on genotoxic stress-activated ABL1/c-Abl. Inhibiting OTULIN or Wnt/β-catenin sensitizes triple-negative breast cancer xenograft tumors to chemotherapeutics and reduces metastasis. Increased OTULIN levels are associated with aggressive molecular subtypes and poor survival in breast cancer patients. Thus, OTULIN-mediated Wnt/β-catenin activation upon genotoxic treatments promotes drug resistance and metastasis in breast cancers.

[1] Department of Radiation Oncology, College of Medicine, University of Tennessee Health Science Center, Memphis, TN 38103, USA. [2] Department of Pathology and Laboratory Medicine, College of Medicine, University of Tennessee Health Science Center, Memphis, TN 38103, USA. [3] Center for Cancer Research, University of Tennessee Health Science Center, Memphis, TN 38103, USA. [4] Department of Medicine, College of Medicine, University of Tennessee Health Science Center, Memphis, TN 38103, USA. [5] Department of Oncology, Fudan University Shanghai Cancer Center, Shanghai, PR China. [6] Department of Breast Surgery, Fudan University Shanghai Cancer Center, Shanghai, PR China. ✉email: zwu6@uthsc.edu

Wnt signaling plays a fundamental role in controlling a myriad of physiological processes involved in embryo development and maintenance of adult stem cells. Dysregulated Wnt signaling and mutation of genes involved in the Wnt signaling cascades are frequently observed in a wide range of human diseases, such as cancer and degenerative diseases[1].

Two major Wnt signaling cascades, β-catenin-dependent canonical Wnt pathway, and β-catenin-independent non-canonical Wnt signaling, have been well characterized. Canonical Wnt signaling is initiated by binding of Wnt ligands (e.g., WNT1, WNT3A, and WNT10B) to heterodimeric receptor complex formed by Frizzled (Fzd) and co-receptor low-density lipoprotein receptor-related protein 6 (LRP6) or its close relative LRP5. The signaling output of the canonical Wnt pathway is determined by the level of cytosolic β-catenin, which is under the strict control of the destruction complex. The destruction complex is composed of AXIN, APC (Adenomatous Polyposis Coli Protein), WTX (Wilms tumor suppressor) and two constitutively active kinases (CK1α/δ and GSK3α/β), which associate with β-catenin and promote its polyubiquitination by phosphorylating the degron motif of β-catenin[2]. Phosphorylated β-catenin can be recognized by F-box/WD-repeat protein β-TrCP within the SCF ubiquitin ligase complex, which facilitates assembly of Lys48 (K48)-linked polyubiquitin chains on β-catenin, leading to its proteasome-dependent degradation[3,4]. Wnt ligand binding with Fzd/LRP receptor complexes together with the recruitment of Dishevelled (Dvl) induces stoichiometric sequestration of AXIN/CK1/GSK3 onto the receptor complex and phosphorylation of LRP5/6. Receptor engagement of the destruction complex to the cell membrane inhibits GSK-3 and blocks the ubiquitination of phosphorylated β-catenin, leading to the accumulation of newly synthesized β-catenin. Increased cytoplasmic β-catenin then translocates into the nucleus where it binds to members of the T cell factor/lymphoid enhancer factor (TCF/LEF) transcription factor family to drive transcription of Wnt/β-catenin target genes such as MYC, CCND1, and AXIN2. Wnt ligands also activate non-canonical Wnt pathways, which are independent of β-catenin[5]. In the planar cell polarity (PCP) pathways, non-canonical Wnt ligands (e.g., WNT5A, WNT5B, and WNT7A) interact with FZDs and co-receptors ROR/RYK that activate the small GTPases Rho and Rac, leading to cytoskeleton rearrangement and cell polarization. Additionally, the Wnt/Ca$^{2+}$ pathway depends on calcium release from the endoplasmic reticulum (ER) upon Wnt binding to FZDs and activation of phospholipase C, which activates protein kinase C and promotes CamKII (Ca$^{2+}$/calmodulin-dependent protein kinase II)-mediated activation of transcription factor nuclear factor of activated T cell (NFAT).

A previous study showed that TCF/LEF H2B-GFP reporter mice for Wnt signaling displayed robust β-catenin activation in hematopoietic stem cells (HSCs) following ionizing radiation (IR)[6]. A canonical Wnt signaling-specific target gene Axin2 was upregulated in mouse intestinal crypt cells after IR[7,8]. In addition, a canonical Wnt/β-catenin gene signature was enriched in Adriamycin-treated mouse and human tumor cells[9]. These findings indicate that DNA damage by genotoxic treatments induces Wnt/β-catenin activation, however, the underlying mechanisms orchestrating β-catenin stabilization and transcriptional activation remain poorly understood.

OTULIN (also known as FAM105B or Gumby) is a deubiquitinase exclusively cleaving polyubiquitin chains linked with linear linkage (Met1/M1-linkage)[10,11]. Recent studies identified several hypomorphic mutations of OTULIN in humans, which lead to autoimmune responses and hyper-inflammation[12–14]. OTULIN was found to interact with the linear ubiquitin assembly complex (LUBAC), which is composed of HOIP (HOIL-1-interacting protein/RNF31), HOIL-1(heme-oxidized IRP2 ubiquitin ligase 1/RBCK1) and Sharpin. As an E3 ligase complex, LUBAC specifically attaches M1-linked polyubiquitin chains on its substrate[15,16]. The OTULIN association with LUBAC is mediated by the interaction between the PUB (peptide: N-glycanase/UBA- or UBX-containing proteins) domain of HOIP and the PIM (PUB-interaction motif) domain of OTULIN[17,18]. Phosphorylation of the highly conserved Tyr56 within the PIM was shown to disrupt the interaction between OTULIN and HOIP, which abrogates OTULIN-dependent suppression of LUBAC and enables LUBAC-regulated signaling. However, the kinase(s) phosphorylating Tyr56 and whether the Tyr56 phosphorylation is a regulated mechanism in cells were unclear[15,19].

Although much has been learned about how LUBAC-assembled linear ubiquitin chains regulate immune signaling such as NF-κB[20], how linear chain disassembly by deubiquitinases like OTULIN influences cellular functions was less understood[12,19]. OTULIN was found to either enhance or suppress NF-κB activation in a context-dependent manner[17,18,21]. In mice harboring inactive Otulin/Gumby mutations (W96R or D336E), Wnt/β-catenin activation was significantly reduced, leading to angiogenic defect and embryonic lethality, whereas the underlying mechanism was largely unexplored[11]. Here, we find that genotoxic stress induces Wnt/β-catenin activation in an OTULIN-dependent manner. DNA damage-induced c-Abl/ABL1 activation, which depends on DNA-PK, is required for Tyr56 phosphorylation of OTULIN. OTULIN phosphorylation at Tyr56 enhances its interaction with β-catenin while diminishing the OTULIN/LUBAC association. Moreover, LUBAC promotes linear ubiquitination of β-catenin in unstimulated cells, which facilitates the proteasomal degradation of β-catenin. Upon genotoxic treatment, increased OTULIN/β-catenin association inhibits β-catenin linear ubiquitination thereby promoting its stabilization and Wnt/β-catenin signaling activation. We further show that increased OTULIN levels are associated with aggressive breast cancer subtypes and correlate with poor survival in breast cancer patients. Increased Wnt/β-catenin activation in TNBC cells upon chemotherapy, which is dependent on OTULIN, facilitates cancer cell adaptation to a drug-resistant state and enhances metastasis. Therefore, targeting genotoxic Wnt/β-catenin activation and OTULIN may mitigate the development of drug resistance and metastasis in TNBC patients receiving chemotherapy.

## Results

**DNA damage induces Wnt/β-catenin activation independent of canonical Wnt receptor complex FZD/LRP.** Previous studies showed that IR induced robust activation of Wnt/β-catenin signaling in mouse hematopoietic stem cells and intestinal crypt cells[6,8]. Enriched canonical Wnt gene signature was also observed in tumor cells treated by adriamycin[9]. We found significant nuclear enrichment of β-catenin in doxorubicin (Dox)-treated MDA-MB-231 cells, supporting the activation of Wnt/β-catenin signaling by genotoxic treatment (Supplementary Fig. 1A). Accordingly, β-catenin levels were substantially increased in MDA-MB-231 cells treated by various genotoxic drugs, including Dox, carboplatin (CBP) and irinotecan (CPT-11), which was accompanied by increased Wnt/β-catenin transactivity measured by TOPFlash reporter assay (Fig. 1a) and upregulated transcription of a canonical Wnt target gene AXIN2 (Supplementary Fig. 1B). Dox treatment-induced Wnt/β-catenin activation was also observed in multiple TNBC cell lines (Fig. 1b). Genotoxic treatments enhanced the levels of active β-catenin (non-phospho β-catenin) and Wnt/β-catenin activation in a time- and dose-dependent manner (Fig. 1c and Supplementary Fig. 1C). The

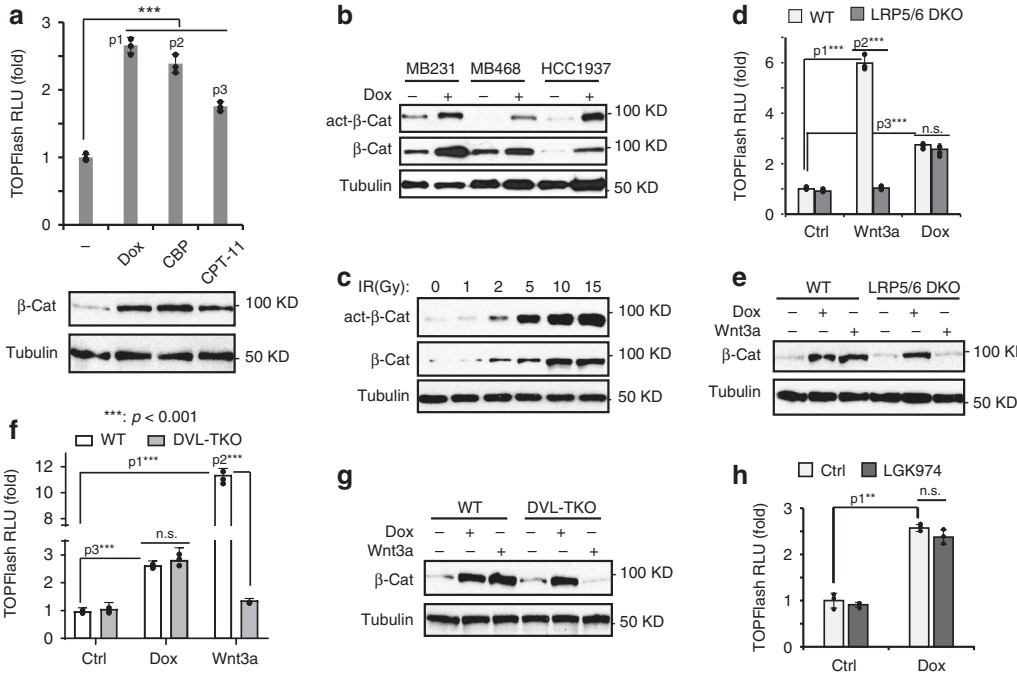

**Fig. 1 DNA damage induces Wnt/β-catenin activation independent of canonical Wnt receptor complex FZD/LRP. a** TOPFlash assay and immunoblotting analysis of MDA-MB-231 cells treated with Dox (2 μg/ml), CBP (10 μg/ml), and CPT-11 (10 μM) for 24 h. $n = 3$ independent experiments. p1 = 2.25E−05, p2 = 8.53E−05, p3 = 0.000118. **b** Immunoblotting analysis of total and active-β-catenin in MDA-MB-231, MDA-MB-468, and HCC1937 cells treated with Dox (2 μg/ml) for 24 h. **c** Immunoblotting analysis of MDA-MB-231 cells exposed to IR with increasing doses. TOPFlash assay (**d**) and immunoblotting analysis of β-catenin expression (**e**) in parental and LRP5/6 DKO HEK293A cells after Wnt3a (20 ng/ml) and Dox (2 μg/ml) treatment for 24 h. $n = 3$ independent experiments in **d**. p1 = 9.93E−06, p2 = 1.09E−05, p3 = 1.39E−05. TOPFlash assay (**f**) and immunoblotting analysis of β-catenin expression (**g**) in parental and DVL1/2/3-TKO HEK293T cells treated as in **d**. $n = 3$ independent experiments in **f**. p1 = 2.95E−06, p2 = 3.21E−06, p3 = 0.000104. **h** TOPFlash assay of MDA-MB-231 cells treated with Dox (2 μg/ml) for 24 h, with or without pretreatment of porcupine inhibitor LGK974 (100 nM). $n = 3$ independent experiments. p1 = 0.000107. The statistical analysis in **a**, **d**, **f** and **h** was performed by two-sided unpaired *t*-test and *p* values are indicated as *$p < 0.05$, **$p < 0.01$, and ***$p < 0.001$. Data are presented as Mean ± SD. Source data are provided as a Source Data file.

activation of Wnt/β-catenin signaling by genotoxic treatment is not limited to TNBC cell lines, as we also observed robust Wnt/β-catenin activation in ER$^+$ MCF7 breast cancer cells, HEK293 cells and a primary TNBC PDX HBrt1071 cells (Supplementary Fig. 1D–H).

The activation of canonical Wnt/β-catenin signaling is initiated by the binding of Wnt ligands with cell membrane-bound heterodimeric FZD/LRP5/6 receptor complex[1]. To our surprise, Dox-induced Wnt/β-catenin activation in cells deficient of both LRP5 and LRP6 was comparable to that in wildtype cells (Fig. 1d, e, Supplementary Fig. 1G), whereas Wnt3a-dependent Wnt/β-catenin activation, as expected, was abrogated in LRP5/6-KO cells[22]. In line with these data, genetic deletion of DVL1/2/3[23], which are required for recruiting destruction complex components AXIN/APC/GSK3 to FZD within the Wnt receptor complex upon activation, abrogated Wnt3a-induced Wnt/β-catenin activation. In contrast, significant activation of Wnt/β-catenin by Dox was observed in cells with DVL1/2/3 deletion (Fig. 1f, g, Supplementary Fig. 1H). We also observed comparable levels of Wnt/β-catenin induction by multiple genotoxic drugs in wildtype, LRP5/6-KO, or DVL-TKO cells (Supplementary Fig. 1I, J). These data suggest that genotoxic agent-induced Wnt/β-catenin activation may be independent of the Wnt ligand binding to the membrane-bound canonical Wnt receptor. Palmitoylation of Wnt proteins is essential for their secretion and effective binding to the FZD proteins, which can be inhibited by porcupine/PORCN inhibitor LGK974[24]. Accordingly, we found that Dox-induced TOPFlash Wnt reporter activity was not diminished by LGK974 (Fig. 1h). Taken together, these data suggested that unlike canonical Wnt/β-catenin signaling, binding of Wnt protein to

membrane-bound FZD/LRP5/6 receptor complex is dispensable for Wnt/β-catenin activation by genotoxic agents.

**OTULIN mediates genotoxic Wnt activation by inhibiting linear ubiquitination.** In mice harboring inactive Otulin/Gumby mutations, Wnt/β-catenin activation was significantly reduced, leading to angiogenic defect and embryonic lethality[11]. We generated OTULIN knockout clones from HEK293 and MDA-MB-231 cell lines which showed increased linear ubiquitination and similar levels of LUBAC subunits compared to wildtype cells (Supplementary Fig. 2A). We found that Wnt/β-catenin activation in OTULIN-depleted MDA-MB-231 cells was suppressed in response to treatment with Dox or Wnt3a (Fig. 2a–c). A similar result was observed in cells exposed to chemodrug CBP (Supplementary Fig. 2B). With TOPFlash reporter assay, we confirmed that the β-catenin transcriptional activity induced by Wnt3A and other genotoxic agents was decreased in OTULIN-KO MDA-MB-231 cells (Fig. 2b). Consistent with the previous report[11], OTULIN overexpression increased the total β-catenin level and Wnt/β-catenin-dependent transcriptional activation (Fig. 2d, e and Supplementary Fig. 2C–E). Moreover, ectopic OTULIN further enhanced Dox- and etoposide (Etop)-induced activation of Wnt/β-catenin in MDA-MB-231 and HEK293T cells, respectively (Supplementary Fig. 2D, E). Interestingly, we observed induction of Wnt/β-catenin activation by OTULIN overexpression in LRP5/6-DKO and DVL-TKO cells (Fig. 2d, e and Supplementary Fig. 2F–I), suggesting that OTULIN promotes β-catenin activation downstream of the Wnt receptor complex.

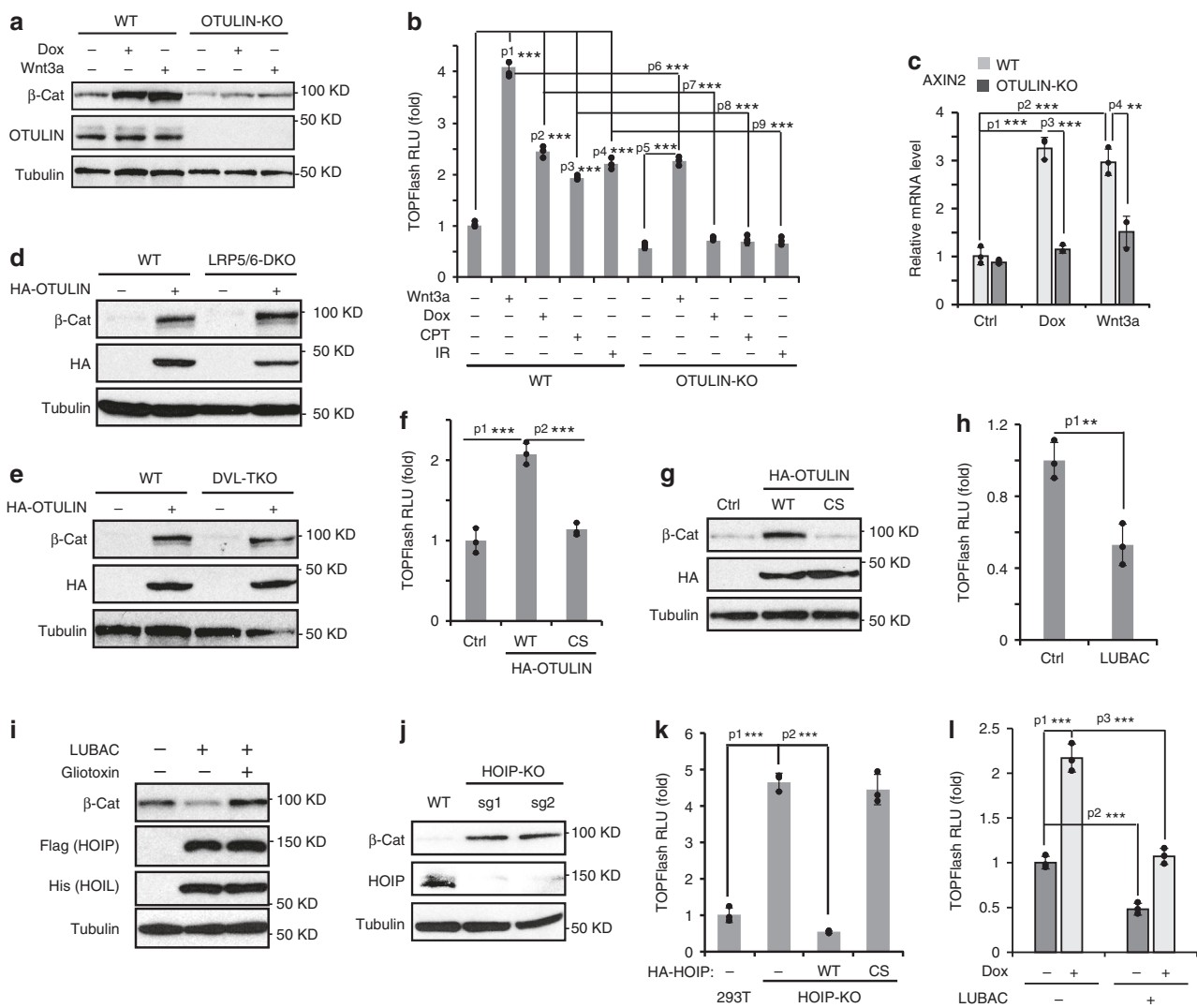

**Fig. 2 OTULIN mediates genotoxic Wnt activation by inhibiting linear ubiquitination. a** Immunoblotting analysis of β-catenin in parental and OTULIN knockout MDA-MB-231 cells treated with Dox (2 μg/ml) or Wnt3a (20 ng/ml) for 24 h. **b** TOPFlash assay of parental and OTULIN knockout MDA-MB-231 cells treated with Wnt3a (20 ng/ml), Dox (2 μg/ml), CPT-11 (10 μM) or IR (10 Gy) for 24 h. $n = 3$ independent experiments. p1 = 4.39E−06, p2 = 3.55E−05, p3 = 1.56E−05, p4 = 5.48E−05, p5 = 8.73E−06, p6 = 5.22E−05, p7 = 1.59E−05, p8 = 1.86E−05, p9 = 3.29E−05. **c** qPCR analysis of *AXIN2* mRNA level in parental and OTULIN knockout MDA-MB-231 cells treated with Dox (2 μg/ml) or Wnt3a (20 ng/ml) for 24 h. $n = 3$ independent experiments. p1 = 0.000188, p2 = 0.000503, p3 = 0.000117, p4 = 0.00413. **d** Immunoblotting analysis of β-catenin expression in parental and LRP5/6-DKO HEK293A cells with or without HA-OTULIN overexpression. **e** Immunoblotting analysis of β-catenin expression in parental and DVL1/2/3-TKO HEK293T cells with or without HA-OTULIN overexpression. TOPFlash assay (**f**) and immunoblotting analysis of β-catenin (**g**) in MDA-MB-231 cells transfected with HA-OTULIN wildtype (WT) or catalytic activity defect C129S mutant (CS). $n = 3$ independent experiments in **f**. p1 = 0.000783, p2 = 0.000497. **h** TOPFlash assay of HEK293T cells transfected with LUBAC components HOIP and HOIL. $n = 3$ independent experiments. p1 = 0.00670. **i** Immunoblotting analysis of β-catenin in MDA-MB-231 cells transfected with HOIP and HOIL and treated with or without LUBAC inhibitor Gliotoxin (1 μM) for 24 h. **j** Immunoblotting analysis of β-catenin expression in parental and HOIP knockout HEK293T cells. **k** TOPFlash assay of parental, HOIP knockout, or HOIP-KO HEK293T cells reconstituted with HOIP WT or C885S mutant. $n = 3$ independent experiments. p1 = 4.06E−05, p2 = 9.49E−06. **l** TOPFlash assay of MDA-MB-231 cells transfected with HOIP and HOIL and treated with Dox (2 μg/ml) for 24 h. $n = 3$ independent experiments. p1 = 0.000297, p2 = 0.000717, p3 = 0.000462. The statistical analysis in **b**, **c**, **f**, **h**, **k** and **l** was performed by two-sided unpaired *t*-test and *p* values are indicated as *$p < 0.05$, **$p < 0.01$, and ***$p < 0.001$. Data are presented as Mean±SD. Source data are provided as a Source Data file.

Since OTULIN is a linear chain-specific deubiquitinase (DUB)[10], and defects in Wnt activation was observed in mice harboring OTULIN inactivating mutations[11], we asked whether OTULIN catalytic activity is essential for mediating Wnt activation by genotoxic agents. Overexpression of OTULIN-WT, but not its catalytic-inactive C129S mutant, enhanced β-catenin transcriptional activity and its protein level in MDA-MB-231 cells (Fig. 2f, g), indicating that removal of M1 ubiquitin chains was required for OTULIN-mediated β-catenin activation. This observation also

suggests that increased linear ubiquitination may suppress Wnt/β-catenin activity. Indeed, we observed that overexpression of the linear ubiquitin ligase LUBAC significantly decreased the β-catenin protein level as well as Wnt/β-catenin transactivity, and inhibiting LUBAC with Gliotoxin[25] substantially restored β-catenin level decreased by LUBAC (Fig. 2h, i and Supplementary Fig. 2J). Moreover, Gliotoxin treatment is sufficient to increase the β-catenin level in HEK293T cells (Supplementary Fig. 2K). In contrast, genetic deletion of HOIP (the catalytic subunit of the

LUBAC ligase) in HEK293T cells, or HOIL-1L in MEFs, increased the β-catenin level and enhanced TOPFlash reporter activity (Fig. 2j, Supplementary Fig. 2L, M), which was attenuated by reconstitution of HOIP-WT but not ligase dead HOIP-CS mutant (Fig. 2k). Importantly, LUBAC overexpression significantly reduced TOPFlash activation in MDA-MB-231 and HEK293T cells treated by Dox or Etop respectively (Fig. 2l and Supplementary Fig. 2N). We also found depletion of CYLD, another DUB which has been shown to cleave linear poly-ubiquitin chain assembled by LUBAC[19,26], had little impact on Dox-induced Wnt/β-catenin activation in MDA-MB-231 cells and MEFs (Supplementary Fig. 2O, P), suggesting an indispensable role of OTULIN in mediating genotoxic Wnt activation. Altogether, these results indicated that OTULIN-dependent inhibition of linear ubiquitination was essential for Wnt/β-catenin activation by genotoxic treatments.

**OTULIN inhibits linear ubiquitination of β-catenin.** Ubiquitination of β-catenin with K48 chains and its subsequent proteasome-dependent degradation keep the canonical Wnt/β-catenin signaling from aberrant activation[5]. Interestingly, we found β-catenin was modified by M1 ubiquitin chains in cells transfected with LUBAC, which was attenuated by co-expression of OTULIN (Fig. 3a). Analyzing polyubiquitin chains conjugated on β-catenin using the UbiCREST assay[27] confirmed that both K48 and M1 chains were conjugated on β-catenin in resting cells (Fig. 3b and Supplementary Fig. 3A). Accordingly, we found linear ubiquitination of β-catenin was significantly decreased in HOIP-KO cells (Fig. 3c and Supplementary Fig. 3B), supporting a critical role of LUBAC in promoting β-catenin linear ubiquitination. We observed that β-catenin interacted with LUBAC in unstimulated cells which was not diminished by Dox treatment (Fig. 3d). In contrast, the interaction of OTULIN with β-catenin was substantially increased upon Dox treatment, which may be responsible for decreased linear ubiquitination and stabilization of β-catenin in cells exposed to genotoxic treatments (Supplementary Fig. 3C, Fig. 3e and h, lanes 1 and 2). Overexpression of OTULIN-WT diminished β-catenin linear ubiquitination which was not affected by ectopic OTULIN-C129S mutant (Supplementary Fig. 3D). We also observed significantly increased β-catenin linear ubiquitination in OTULIN-KO cells compared to that in the wildtype cells (Fig. 3e), whereas this increased β-catenin linear ubiquitination was suppressed by reconstitution of OTULIN-WT, but not -C129S mutant, in OTULIN-deficient cells (Fig. 3f). Consistently, decreased β-catenin level in OTULIN-KO cells was rescued only by wildtype OTULIN (Fig. 3f, right panel). These data suggest that OTULIN-dependent disassembly of the M1 chains on β-catenin may be required for β-catenin activation by genotoxic agents.

Consistent with a previous study showing that Lys 19 (K19) and Lys 49 (K49) of β-catenin were required for its ubiquitination[28], we found conjugation of K48 chains to β-catenin was mostly abrogated in β-catenin K19/49 R mutant (Supplementary Fig. 3E). However, mutation of K19, K49, or both minimally affected linear ubiquitination of β-catenin by LUBAC (Supplementary Fig. 3F). To identify the lysine(s) required for linear chain conjugation on β-catenin, we individually mutated 17 lysine residues within β-catenin which have been identified for ubiquitination in PhosphoSitePlus database (www.phosphosite.org). We found that K133R mutation substantially decreased the linear ubiquitination of β-catenin compared with other mutants (Fig. 3g and Supplementary Fig. 3G), suggesting that K133 is required for linear ubiquitination of β-catenin. In vitro ubiquitination analyses also confirmed that K133 is essential for LUBAC-mediated β-catenin ubiquitination (Supplementary Fig. 3H). To our surprise, we found that K133R mutation also dramatically decreased the K48 ubiquitination of β-catenin, which was comparable to the K19/49 R mutation (Fig. 3g, bottom panel). However, the abundance of linear chain remained unchanged in K19R, K49R, and K19/49 R mutants, suggesting that linear ubiquitination of β-catenin may occur prior to K48 chain conjugation. This notion is in accordance with our observation that the K48 ubiquitination of β-catenin also was increased along with the linear ubiquitination in OTULIN-KO cells, which was diminished by reconstitution of wildtype OTULIN (Fig. 3f, bottom panel). These results led us to speculate that linear ubiquitination of β-catenin may modulate its subsequent conjugation with the K48-polyubiquitin chains. Accordingly, HOIP depletion or treatment with Dox or LUBAC inhibitor Gliotoxin reduced both linear and K48 ubiquitination of β-catenin, which correlated with increased levels of β-catenin (Supplementary Fig. 3B and Fig. 3h). Moreover, LUBAC-dependent β-catenin decrease was blocked by a proteasome inhibitor (Supplementary Fig. 3I), suggesting that proteasome-dependent β-catenin degradation is critical for β-catenin suppression by LUBAC.

LUBAC-catalyzed linear ubiquitination generally plays critical roles in regulating signaling transduction rather than promoting protein degradation[16]. SCFβ-TrCP E3 ubiquitin ligase-mediated K48 ubiquitination is essential for proteasomal degradation of β-catenin[1]. LUBAC overexpression enhanced both linear and K48 ubiquitination of β-catenin in HEK293T cells, whereas co-expression of a dominant-negative β-TrCP F-box-deletion mutant selectively suppressed K48 ubiquitination of β-catenin promoted by LUBAC (Fig. 3i), suggesting that β-TrCP is required for increased β-catenin K48 ubiquitination by LUBAC. In accordance, β-catenin with K19/49 R mutations, which block its K48 ubiquitination, was resistant to LUBAC-promoted degradation (Supplementary Fig. 3J). We found the degron phosphorylation of β-catenin was dispensable for its linear ubiquitination and Dox treatment effectively decreased linear ubiquitination of a β-catenin degron SA mutant (S33A/S37A/T41A/S45A) (Supplementary Fig. 3K). Inhibiting GSK3 with CHIR99021 stabilizes β-catenin by diminishing degron phosphorylation. The β-catenin stabilization by CHIR99021 was substantially decreased in OTULIN-KO cells, compared to that in WT cells (Supplementary Fig. 3L), indicating that GSK3-dependent phosphorylation and LUBAC-mediated linear ubiquitination may parallelly contribute to the effective binding of β-catenin with β-TrCP and subsequent degradation. Consistently, the stability of the K133R mutant is comparable to the K19/49 R or the SA mutants of β-catenin which both are defective of β-TrCP-mediated K48 ubiquitination (Supplementary Fig. 3M). Intriguingly, we found inhibiting LUBAC with Gliotoxin decreased the interaction between β-catenin and β-TrCP (Fig. 3j), indicating that linear ubiquitination of β-catenin may enhance its interaction with β-TrCP. The WD40-repeat of β-TrCP is required for its binding to phosphorylated degron of β-catenin[29]. We confirmed that the WD40-repeat domain is essential for the interaction between β-catenin and β-TrCP in response to Dox treatment (Supplementary Fig. 3N). Although β-catenin degron phosphorylation was not affected by K133R mutation, the interaction between β-catenin K133R and β-TrCP was decreased (Fig. 3k), suggesting that the linear ubiquitination defect may be responsible for the decreased interaction. Interestingly, WD40-repeats have been shown to bind ubiquitin chains[30]. Using an in vitro pull-down assay, we found the linear tetra-Ub proteins could interact with full-length β-TrCP but not the WD40-deletion mutant in cell lysates (Supplementary Fig. 3O). These data suggest that genotoxic treatment may decrease linear ubiquitination of β-catenin through promoting OTULIN association with β-catenin,

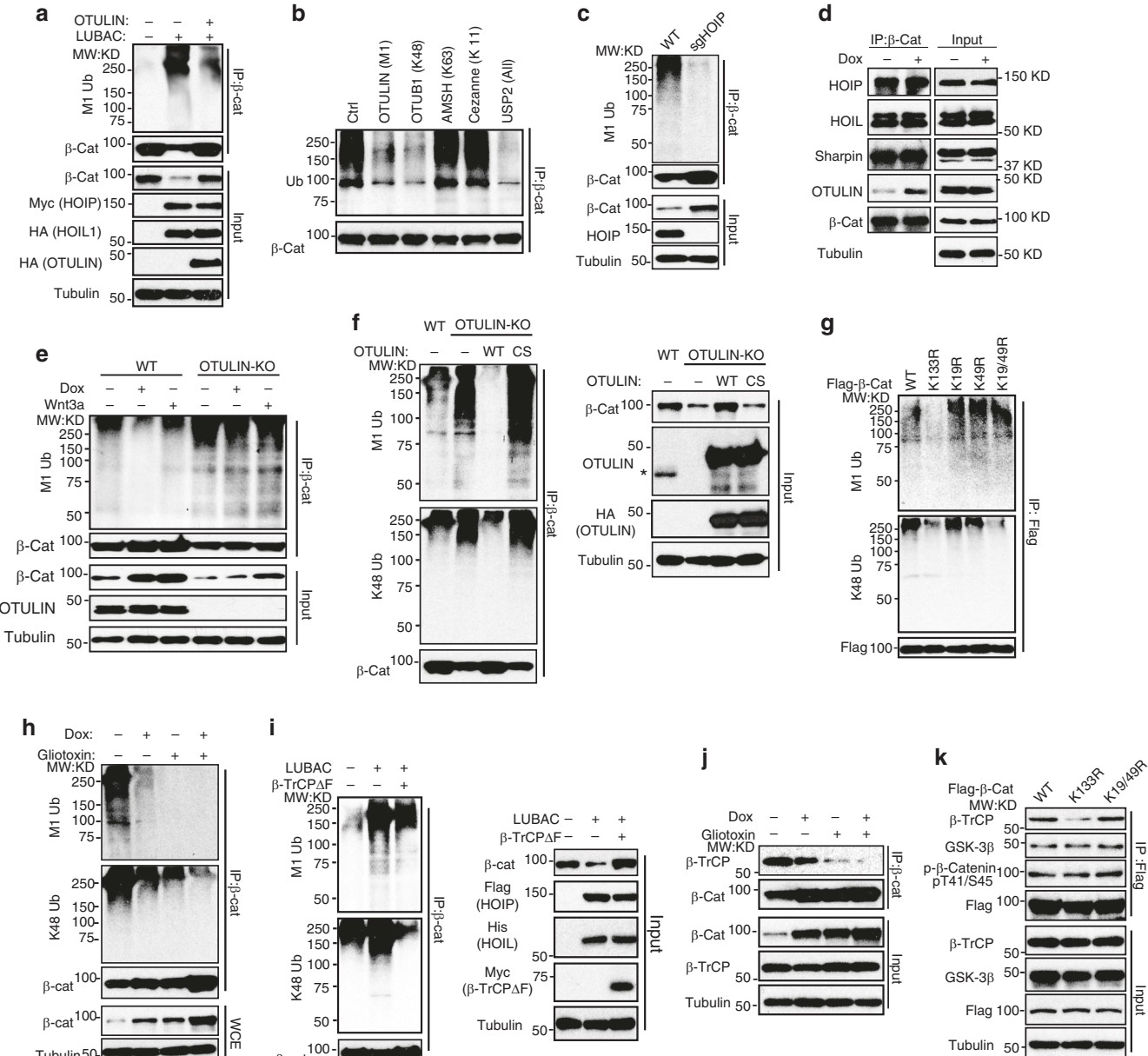

**Fig. 3 OTULIN inhibits linear ubiquitination of β-catenin. a** M1-linked ubiquitination of immunoprecipitated β-catenin from MDA-MB-231 cells transfected with LUBAC (HOIP + HOIL1) or along with HA-OTULIN. **b** UbiCREST analysis of β-catenin ubiquitination. MDA-MB-231 cells were treated with MG132 (10 μM) for 16 h. Then, β-Catenin was immunoprecipitated and incubated with indicated DUBs. The ubiquitination of β-Catenin precipitates after DUB digestion was analyzed by Western blot. **c** Linear ubiquitination of immunoprecipitated β-catenin from parental or HOIP knockout HEK293T cells. **d** Immunoblotting of co-immunoprecipitated proteins with β-Catenin in MDA-MB-231 cells treated with Dox (2 μg/ml) and MG132 (10 μM) for 8 h. **e** Linear ubiquitination of β-Catenin immunoprecipitates from parental and OTULIN knockout MDA-MB-231 cells treated as indicated. **f** Levels of linear Ub chain and K48 Ub chain in β-Catenin immunoprecipitates from parental, OTULIN knockout MDA-MB-231 cells or OTULIN-KO cells reconstituted with OTULIN-WT or –C129S mutant. *: endogenous OTULIN. **g** Linear and K48 ubiquitination of Flag-β-catenin WT, K133R, K19R, K49R, or K19/49R mutants in HEK293T cells. **h** Linear and K48 ubiquitination of β-Catenin immunoprecipitates in MDA-MB-231 cells treated with Dox (2 μg/ml), Gliotoxin (1 μM) or in combination for 24 h. **i** Linear and K48 ubiquitination of β-Catenin immunoprecipitates in HEK293T cells transfected with LUBAC and F-box-deleted β-TrCP mutant. **j** The analysis of β-catenin and β-TrCP interaction in MDA-MB-231 cells treated as in **h**. **k** Co-IP assay with anti-Flag antibody in HEK293T cells transfected with Flag-β-catenin WT, K133R, or K19/49 R. Source data are provided as a Source Data file.

leading to reduced β-TrCP binding, decreased K48 ubiquitination, and stabilization of β-catenin.

**OTULIN Tyr56 phosphorylation upon DNA damage promotes its association with β-catenin.** OTULIN forms a complex with LUBAC through HOIP in unstimulated cells[17,18,31]. We found genotoxic treatment enhanced the interaction between OTULIN and β-catenin. Meanwhile, the association between OTULIN and

LUBAC was dramatically decreased (Fig. 4a, b and Supplementary Fig. 4A). Further characterization of domains required for the interaction between OTULIN and β-catenin revealed that the armadillo-repeats of β-catenin and the N-terminal region of OTULIN are required for their interaction (Fig. 4c, d and Supplementary Fig. 4B). Increased HOIP expression did not significantly reduce OTULIN association with β-catenin upon Dox treatment (Supplementary Fig. 4C), suggesting that LUBAC binds

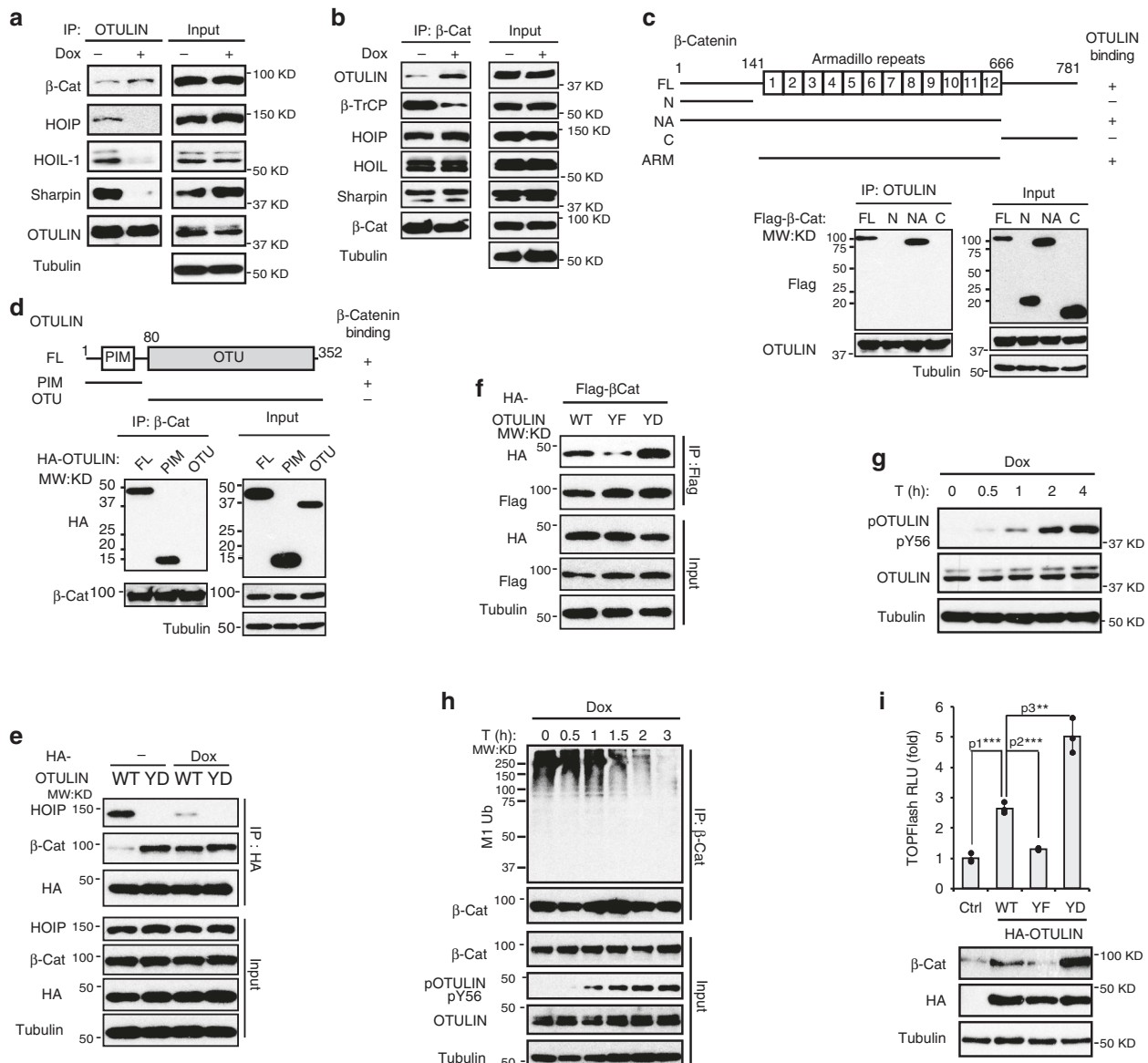

**Fig. 4 OTULIN Tyr56 phosphorylation upon DNA damage promotes its association with β-catenin. a** Co-IP assay with anti-OTULIN antibody in MDA-MB-231 cells treated by Dox (2 μg/ml) and MG132 (10 μM) for 8 h. **b** Co-IP assay with anti-β-Catenin antibody in MDA-MB-231 cells treated as in **a**. **c** Mapping of β-Catenin domain required for interacting with OTULIN. **d** Mapping of OTULIN domain required for interacting with β-Catenin. **e** MDA-MB-231 cells transfected with OTULIN WT or Y56D mutant were treated with MG132 (10 μM) for 8 h, then with MG132 in combination of Dox (2 μg/ml) for 8 h. **f** The Co-IP analysis of Flag-β-Catenin and HA-OTULIN WT, Y56F or Y56D in HEK293T cells. MG132 (10 μM) was added to the culture 16 h before harvest. **g** Detection of OTULIN phosphorylation at Tyr56 in MDA-MB-231 cells treated with Dox (2 μg/ml) for times indicated. **h** Linear ubiquitination of precipitated β-Catenin from MDA-MB-231 cells treated with Dox (2 μg/ml) for times as indicated. **i** TOPFlash assay in MDA-MB-231 cells transfected with OTULIN WT, Y56F, or Y56D. $n = 3$ independent experiments. p1 = 0.000300, p2 = 0.000286, p3 = 0.00247. The statistical analysis in **i** was performed by two-sided unpaired $t$-test and $p$ value is indicated as *$p < 0.05$, **$p < 0.01$, and ***$p < 0.001$. Data are presented as Mean ± SD. Source data are provided as a Source Data file.

to β-catenin at a distinct motif without competing with OTULIN. The OTULIN N-terminal PIM domain is essential for its interaction with HOIP, and phosphorylation of Tyr56 within the OTULIN PIM domain could disrupt the interaction between the OTULIN and HOIP[17,18]. We found Dox treatment significantly decreased the OTULIN association with HOIP (Fig. 4e). Moreover, a phospho-mimetic mutation on Tyr56 (Y56D) also abolished the association between OTULIN and HOIP, suggesting that genotoxic treatment may induce OTULIN phosphorylation at Tyr56, which disrupts the OTULIN-HOIP association. However, while the Y56D mutation decreased OTULIN interaction with HOIP, it enhanced OTULIN association with β-catenin

(Fig. 4e, f and Supplementary Fig. 4B), suggesting that OTULIN Tyr56 phosphorylation may promote OTULIN interaction with β-catenin. Consistently, we found OTULIN Tyrosine phosphorylation at Y56 was induced upon genotoxic treatments (Supplementary Fig. 4D), which was correlated with the increased association between OTULIN and β-catenin (Fig. 4b).

To further characterize the OTULIN phosphorylation upon genotoxic treatments, we developed an antibody specifically recognizing Tyr56 phosphorylation of OTULIN (4E−S4G). This antibody detected OTULIN phosphorylation in cells exposed to genotoxic drugs, which was abrogated by phosphatase treatment or Tyr56 mutation. Tyr56 phosphorylation was not observed in

cells treated with Wnt3a (Supplementary Fig. 4H), suggesting that this phosphorylation is specifically induced in genotoxic stress-induced Wnt/β-catenin activation. Dox treatment induced a time-dependent OTULIN Tyr56 phosphorylation in MDA-MB 231 cells (Fig. 4g). Accordingly, we observed a decrease in linear ubiquitination of β-catenin with comparable kinetics in cells treated with Dox, which correlated with the increase of OTULIN phosphorylation (Fig. 4h). Wnt3a treatment induced a decrease of β-catenin linear ubiquitination with much-delayed kinetics (Supplementary Fig. 4I), which may be due to, at least in part, the lack of OTULIN phosphorylation. In parallel, the decreased OTULIN association with HOIP correlated with increased OTULIN phosphorylation at Tyr56 upon genotoxic treatment (Supplementary Fig. 4J). Interestingly, OTULIN-Y56D mutant overexpression induced more robust Wnt/β-catenin activation (Fig. 4i), which may be a consequence of the increased interaction between β-catenin and OTULIN-Y56D mutant (Fig. 4f). Taken together, these data support that OTULIN phosphorylation at Tyr56 upon DNA damage enhances its interaction with β-catenin while disrupting OTULIN/HOIP association, resulting in increased β-catenin stability and Wnt/β-catenin-dependent transcriptional upregulation.

**DNA damage-activated c-Abl is required for OTULIN phosphorylation.** To identify the kinase that is responsible for the OTULIN Tyr56 phosphorylation in response to DNA damage, we performed a siRNA screen with a tyrosine kinase sub-library, which consists of 88 tyrosine kinases. We found knockdown of ABL1 significantly reduced OTULIN Tyr56 phosphorylation by Dox (Fig. 5a). With a complimentary screen with a variety of tyrosine kinase inhibitors, we confirmed that inhibiting ABL1/c-Abl with dasatinib abrogated OTULIN Tyr56 phosphorylation in response to Dox (Supplementary Fig. 5A). This inhibition was also observed in cells treated with imatinib (Fig. 5b). ITK (Interleukin-2-Inducible T-cell Kinase) depletion also inhibited Tyr56 phosphorylation in our screen. While inhibiting c-Abl abrogated β-catenin activation by Dox treatment (Supplementary Fig. 5B), little effect on Wnt/β-catenin activation was observed in cells treated with Dox in combination with ITK inhibitor BMS 509744 (Supplementary Fig. 5C), which prompted us to focus on the role of c-Abl in modulating genotoxic Wnt signaling. Inhibiting c-Abl with imatinib also abrogated OTULIN Tyr56 phosphorylation in multiple TNBC cell lines and HBrt1071 PDX cells (Supplementary Fig. 5D). Consistently, ABL1 deletion in MDA-MB-231 cells suppressed OTULIN Tyr56 phosphorylation by Dox, CBP, and CPT-11 (Supplementary Fig. 5E). This c-Abl-dependent Tyr56 phosphorylation was also observed in HEK293 cells treated with Etop (Supplementary Fig. 5F). Previous studies have shown that c-Abl was activated in response to genotoxic stress which promotes its participation in DNA damage response[32,33]. We found Dox treatment induced a robust c-Abl activation measured by its increased kinase activity (Fig. 5c). Consistently, phosphorylation of OTULIN at Tyr56 by c-Abl was also significantly increased in a time-dependent manner upon Dox treatment (Fig. 5c and Supplementary Fig. 5G). We observed OTULIN phosphorylation in the cytoplasm along with increased cytoplasmic c-Abl level at later time points after Dox treatment, suggesting that c-Abl may translocate into the cytoplasm after its nuclear activation upon genotoxic stress and promote OTULIN phosphorylation (Supplementary Fig. 5H and S5I). Increased association between OTULIN and c-Abl was also detected in Dox-treated cells (Fig. 5d), which depended on the C-terminal OTU domain of the OTULIN (Supplementary Fig. 5J). Importantly, depleting ABL1 in MDA-MB-231 cells abolished Dox-induced OTULIN Tyr56 phosphorylation, which was rescued by ABL1-WT but not kinase-dead K290R mutant (Fig. 5e). These data suggest that c-Abl is activated in response to genotoxic treatment, which may be responsible for Tyr56 phosphorylation of OTULIN.

Along with the increased OTULIN Tyr56 phosphorylation, β-catenin was also increased in cells treated with Dox, which was suppressed by c-Abl inhibitors or genetic deletion of ABL1 (Fig. 5b, e). Accordingly, the induction of Wnt/β-catenin-target gene AXIN2 transcription by Dox was also abrogated by inhibiting c-Abl (Fig. 5f). In contrast, inhibiting c-Abl did not affect Wnt3a-induced β-catenin stabilization (Supplementary Fig. 5B, bottom panel), suggesting that c-Abl is specifically required for DNA damage-induced Wnt/β-catenin activation through mediating OTULIN Tyr56 phosphorylation.

Previous studies have shown that DNA damage response pivotal kinases ATM and DNA-PK can phosphorylate c-Abl in response to DNA damage which upregulates c-Abl activity[34,35]. We found OTULIN Tyr56 phosphorylation in HEK293T cells and MDA-MB-231 cells, treated with Etop or Dox respectively, was abolished by DNA-PK inhibitor Nu7441, but not inhibitors of ATM (Ku55933), ATR (VE-821) or IKK (TPCA) (Supplementary Fig. 5K). Consistently, DNA-PKcs knockdown remarkably reduced OTULIN Tyr56 phosphorylation in cells treated with genotoxic drugs (Fig. 5g). This observation was further confirmed by our in vitro c-Abl kinase assay showing that Dox-induced OTULIN phosphorylation was suppressed by inhibiting DNA-PK, which was comparable to that in cells treated with imatinib (Supplementary Fig. 5L). Moreover, inhibiting DNA-PK, but not ATM, attenuated the decrease of β-catenin linear ubiquitination in response to Dox treatment (Fig. 5h), as well as suppressed DNA damage-induced Wnt/β-catenin activation (Supplementary Fig. 5M). We also found DNA-PK activity is dispensable for Wnt3a-induced canonical Wnt/β-catenin activation (Supplementary Fig. 5N). Along with the observation that c-Abl activation is not required for this canonical Wnt signaling pathway (Supplementary Fig. 5B), our data suggest that DNA-PK-dependent c-Abl activation is specifically required for DNA damage-induced Wnt/β-catenin activation. Inhibiting either c-Abl or DNA-PK with imatinib or Nu-7441 blocked Dox-enhanced OTULIN interaction with β-catenin while stabilizing OTULIN/HOIP interaction (Fig. 5i). We observed similar results in ABL1-KO cells upon Dox treatment (Supplementary Fig. 5O). All these results indicate that, upon DNA damage, DNA-PK-activated c-Abl phosphorylates OTULIN at Tyr56, which promotes OTULIN association with β-catenin, resulting in β-catenin stabilization and Wnt/β-catenin signaling activation.

DNA damage also induces activation of NF-κB and MAPK pathways which could modulate cell death in response to genotoxic stress[36,37]. OTULIN has been shown to play a critical role in regulating MAPK and cell death pathways[19,21]. While TNFα-induced activation of NF-κB and MAPK pathways was not significantly affected by ABL1-deficiency, Dox-induced NF-κB activation was attenuated in ABL1-KO cells (Supplementary Fig. 5P). Accordingly, increased Capasese3 activation by Dox was observed in ABL1-KO cells compared to WT cells (Supplementary Fig. 5Q). These data suggest that ABL1 may be required for efficient NF-κB activation upon genotoxic stress. The potential role of c-Abl-dependent OTULIN phosphorylation in regulating genotoxic NF-κB signaling warrant further investigation.

**Genotoxic Wnt activation promotes drug resistance and metastasis in breast cancer.** Mutations of genes involved in the Wnt/β-catenin pathway have been observed in various cancers. Wnt signaling plays critical roles in maintaining cancer stem cells (CSCs) and promoting EMT, which both contribute significantly

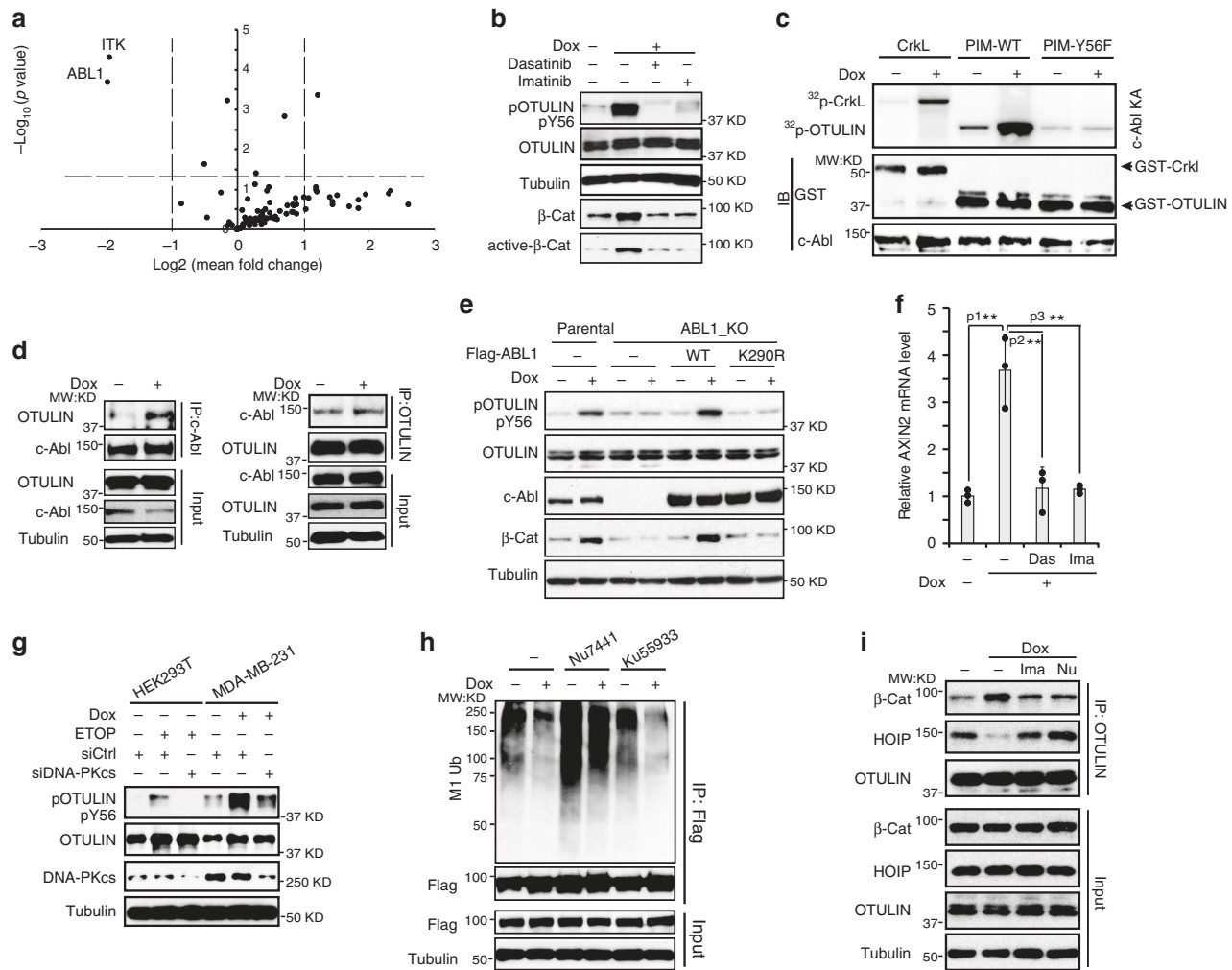

**Fig. 5 DNA damage-activated c-Abl is required for OTULIN phosphorylation. a** Tyrosine kinase siRNA sublibrary was used to screen the kinase responsible for OTULIN Tyr56 phosphorylation. The pooled results from three replicates were presented as a volcano plot. n = 3 independent experiments. **b** Detection of OTULIN phosphorylation at Tyr56 in MDA-MB-231 cells pretreated with Dasatinib (10 μM) or Imatinib (10 μM) and subsequently treated with Dox (2 μg/ml) for 90 min. **c** c-Abl kinase assay in MDA-MB-231 cells treated with Dox (2 μg/ml) for 90 min, using recombinant CrkL, OTULIN-WT or -Y56F mutant PIM fragment as substrates. **d** Co-IP analysis of c-Abl and OTULIN in MDA-MB-231 cells treated with Dox (2 μg/ml) for 90 min. **e** Detection of OTULIN phosphorylation at Tyr56 in parental, ABL1 knockout MDA-MB-231 cells, and ABL1-KO cells transfected with ABL1-WT or K290R mutant after Dox (2 μg/ml) treatment for 90 min. **f** qPCR analysis of AXIN2 mRNA level in MDA-MB-231 cells pretreated with Dasatinib (10 μM) or Imatinib (10 μM) and subsequently treated with Dox (2 μg/ml) for 24 h. n = 3 independent experiments. Data are presented as Mean±SD, p1 = 0.00388, p2 = 0.00788, p3 = 0.00455. **g** Immunoblotting analysis of HEK293T and MDA-MB-231 cells, with or without DNA-PKcs knockdown, treated with Etoposide (10 μM) and Dox (2 μg/ml) for 90 min, respectively. **h** Linear ubiquitination in Flag-β-Catenin-immunoprecipitates from MDA-MB-231 cells pretreated with DNA-PK inhibitor Nu7441 (10 μM) or ATM inhibitor Ku55933 (10 μM) and subsequently treated with MG132 (10 μM) and Dox (2 μg/ml) for 8 h. **i** Co-IP analysis of OTULIN, β-Catenin and HOIP in MDA-MB-231 cells pretreated with Imatinib (10 μM) or Nu7441 (10 μM) and subsequently treated with MG132 (10 μM) and Dox (2 μg/ml) for 8 h. The statistical analysis in **f** was performed by two-sided unpaired t-test and p value is indicated as \*p < 0.05, \*\*p < 0.01, and \*\*\*p < 0.001. Source data are provided as a Source Data file.

to cancer metastasis and drug resistance[38]. We found stably expressing OTULIN in MMTV-PyMT mammary epithelial cells from FVB mice (PyMT cells) significantly increased β-catenin level as well as *Axin2* transcription, compared with the control PyMT cells (Supplementary Fig. 6A and S6B). The increased Wnt/β-catenin activation in OTULIN-stable PyMT cells was correlated with substantially enhanced drug resistance to treatment with Dox (~7.5-fold increase in IC50) or CPT-11 (~6.2-fold increase in IC50) (Fig. 6a and Supplementary Fig. 6C). In contrast, the deletion of OTULIN in MDA-MB-231 cells substantially decreased (4.5-fold) the IC50 of Dox compared to the parental cells (Supplementary Fig. 6D). We also found that c-Abl, the upstream kinase modulating OTULIN-mediated genotoxic Wnt activation, may also enhance drug resistance in cancer cells.

ABL1-KO MDA-MB-231 cells were significantly more sensitive to Dox treatment compared to the wildtype cells (Fig. 6b). This observation was further corroborated by our finding that inhibiting c-Abl with imatinib significantly sensitizes MDA-MB-231 cells to Dox or CBP treatment (Supplementary Fig. 6E). These data support that c-Abl/OTULIN-dependent genotoxic Wnt/β-catenin activation may promote drug resistance in TNBC cells.

To further validate the role of OTULIN-mediated Wnt activation in promoting breast cancer drug resistance in vivo, we established orthotopic TNBC xenograft models with WT or OTULIN-KO MDA-MB-231 LM2 cells. OTULIN deletion showed little effect in xenograft tumor growth, but it significantly sensitized MDA-MB-231 xenografts to Dox treatment and remarkably suppressed tumor growth (Fig. 6c, d). Moreover,

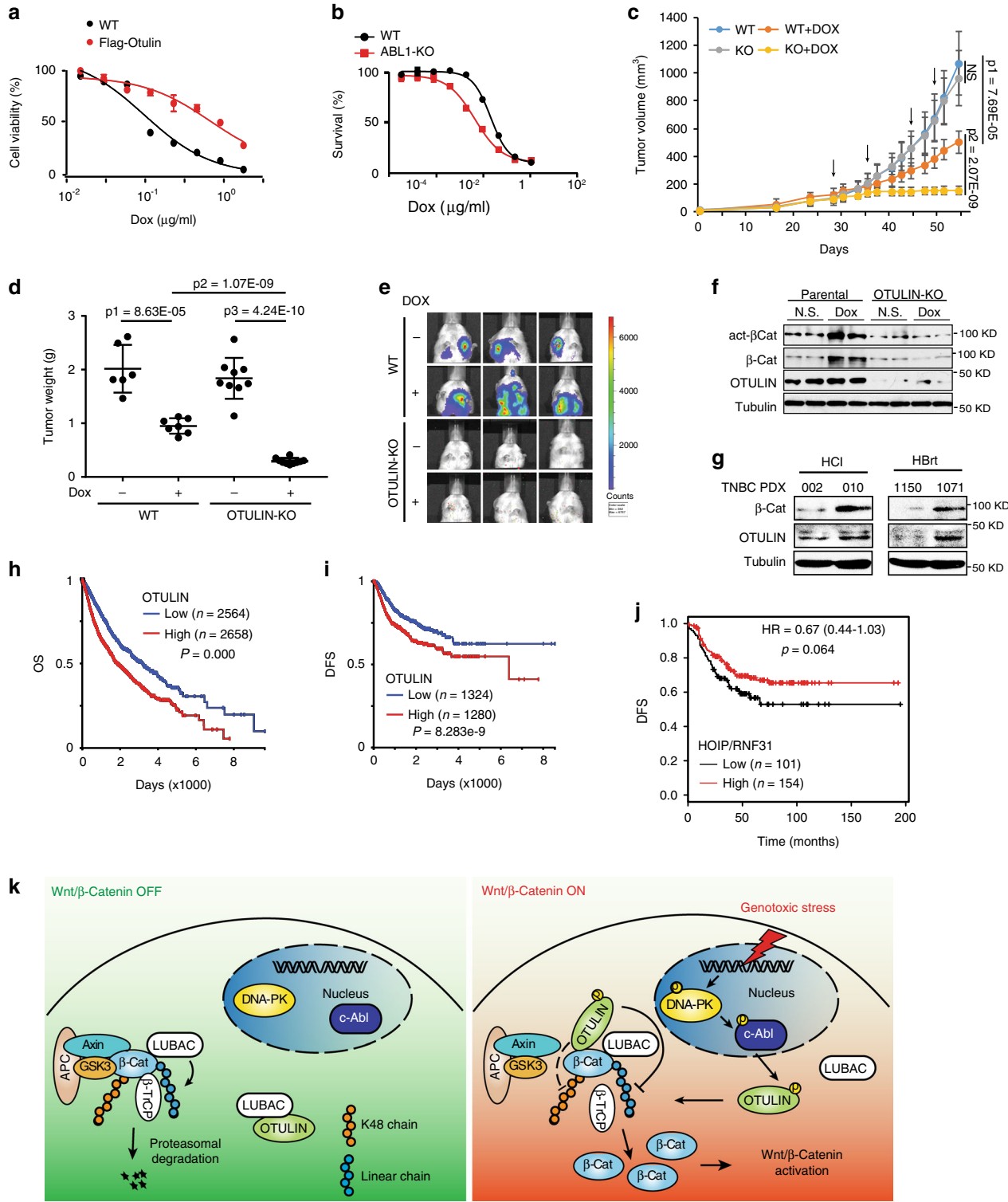

OTULIN-deletion also substantially reduced lung metastasis of the MDA-MB-231 LM2 xenografts with or without Dox treatment (Fig. 6e). Consistently, Dox-induced β-catenin activation was abrogated in OTULIN-KO xenograft tumors (Fig. 6f and Supplementary Fig. 6F), suggesting a potential mechanistic link between OTULIN-mediated genotoxic Wnt/β-catenin activation and increased drug resistance and metastasis in TNBC. In accordance, increased drug sensitivity in OTULIN-KO MDA-MB-231 cells was mitigated by the reconstitution of constitutively active β-catenin SA mutant or OTULIN-WT (Supplementary

Fig. 6G), supporting an essential role of Wnt/β-catenin activation mediated by OTULIN in promoting drug resistance in TNBC cells exposed to genotoxic treatments.

To examine if Wnt/β-catenin activity is essential for metastasis and drug resistance in TNBC cells, we inhibited Wnt/β-catenin-mediated transcription by β-catenin inhibitor ICG-001 as reported[39]. ICG-001 treatment significantly reduced basal and OTULIN-enhanced MDA-MB-231 cell migration (Supplementary Fig. 6H). Moreover, treating MDA-MB-231 LM2 xenograft tumors with Dox in combination with ICG-001 also further

**Fig. 6 Genotoxic Wnt activation promotes drug resistance and metastasis in breast cancer. a** Cell viability of parental and PyMT cells stably expressing OTULIN at 72 h after Dox treatment. Dox $IC_{50}$ is 0.1 μg/ml and 0.75 μg/ml in parental and Flag-OTULIN PyMT cells, respectively. $n = 3$ independent experiments. **b** Cell viability of parental and ABL1 knockout MDA-MB-231 cells in response to Dox treatment for 72 h. Dox $IC_{50}$ is 0.063 μg/ml and 0.016 μg/ml for parental and ABL1 knockout MDA-MB-231 cells, respectively. $n = 3$ independent experiments. **c**–**f** Orthotopic xenograft model generated with parental or OTULIN-KO LM2 cells were treated with Dox (1.5 mg/kg/wk). Tumor growth curves were shown in **c**. p1 = 7.69E−05, p2 = 2.07E−09. At the endpoint, tumors were dissected and weighed as shown in **d**. p1 = 8.63E−05, p2 = 1.07E−09, p3 = 4.24E−10. Lung metastasis was monitored by bioluminescent imaging (**e**). Tumor samples were examined by Western blot for indicated proteins (**f**). $n = 5$ mice/group. **g** Immunoblotting analysis of β-Catenin and OTULIN expression in TNBC PDX cells, treatment naïve: HCI-002 and HBrt1150; chemorefractory: HCI-010 and HBrt1071. Overall survival (**h**) and disease-free survival (**i**) analysis in breast cancer patients from a combined dataset (TCGA-TARGET-GTEx). **j** Disease-free survival analysis in TNBC patients who received systemic chemotherapy stratified on the levels of HOIP (KM-Plot, two-sided, no adjustment for multiple comparisons). **k** Graphic summary of OTULIN-mediated genotoxic Wnt signaling pathway. The statistical analysis in **c** and **d** was performed by two-sided unpaired *t*-test and in **h**–**j** was performed by the two-sided Log-rank test. *p* values are shown as \**p* < 0.05, \*\**p* < 0.01, and \*\*\**p* < 0.001. Data are presented as Mean ± SD. Source data are provided as a Source Data file.

reduced tumor growth compared with Dox treatment alone (Supplementary Fig. 6I, J). These results were in line with the increased OTULIN and β-catenin levels we observed in chemo-refractory PDX samples (HCI-010 and HBrt1071), compared with treatment-naïve PDX samples (HCI-002 and HBrt1150) from TNBC patients (Fig. 6g), supporting a critical role of OTULIN-mediated Wnt/β-catenin activation in drug resistance developed in TNBC patients.

We further examined the OTULIN levels in breast tumor samples and normal breast tissues. OTULIN levels were significantly increased in tumor samples compared with that in normal breast tissues (Supplementary Fig. 6K). Analyzing OTULIN transcription levels in TCGA-BRCA genomic dataset revealed that OTULIN expression is significantly higher in the basal-like subtype of breast cancer than that in other molecular subtypes (Supplementary Fig. 6L). We further focused on the TNBC patients included in the TCGA dataset, which were further characterized into six molecular subtypes[40]. OTULIN levels were higher in Basal-like 2 (BL2), Immunomodulatory (IM), and Mesenchymal-like (M) subtypes compared with the others (Supplementary Fig. 6M). Interestingly, upregulated Wnt signaling gene signatures were found to be enriched in BL2 and M subtypes[40], indicating a potential correlation between increased OTULIN level and enhanced Wnt target gene expression. To further validate this correlation, we retrieved a list of genes whose expression significantly correlated with OTULIN levels in breast cancer patients within the TCGA-PanCAN study[41]. By GSEA analysis of the genes positively correlated to OTULIN levels (Spearman's $R \geq 0.4$, $p < 1.69E-39$), we found genes signatures such as CTNNB1_TARGETS and MYCN_TARGETS_WITH_E-BOX, along with Basal-like breast cancer subtype genes signatures, were enriched (Supplementary Fig. 6N). Accordingly, Genes downregulated by Wnt activation (GSE26351_UNSTIM_VS_WNT_PATHWAY_STIM_HEMATOPOIETIC_PROGENITORS) was enriched in OTULIN-negatively correlated genes (Spearman's $R \leq -0.4$, $p < 1.69E-39$) in breast cancer patients (Supplementary Fig. 6O). All these data support that increased OTULIN levels strongly correlated with upregulated Wnt/β-catenin signaling and TNBC/basal-like molecular subtypes in breast cancer patients.

We further analyzed the survival interval of breast cancer patients based on their expression levels of OTULIN. By extracting breast cancer patient data from the combined dataset (TCGA-TARGET-GTEx), we found patients with high OTULIN levels ($n = 2564$) had significantly shorter overall survival interval than those with lower OTULIN expression ($n = 2658$) (Fig. 6h). Consistently, significantly decreased disease-free survival was found in patients with increased OTULIN levels ($n = 1280$) compared to those with lower OTULIN expression ($n = 1324$) (Fig. 6i). This observation was further corroborated in two

independent breast cancer datasets, in which higher OTULIN levels significantly correlated with short disease-free survival in basal breast cancer patients (GSE21653) (Supplementary Fig. 6P), and poor distant metastasis-free survival in breast cancer patients received chemotherapies (KM plotter, Supplementary Fig. 6Q)[42]. These results indicate that increased OTULIN levels in basal-like/TNBC patients significantly associated with poor prognosis, short survival, and increased drug resistance, which may be dependent on OTLIN-promoted Wnt/β-catenin activation. Overexpressing LUBAC catalytic subunit HOIP significantly attenuated OTULIN-dependent drug resistance in MDA-MB-231 cells (Supplementary Fig. 6R). Accordingly, breast cancer patients with high levels of LUBAC subunit HOIP or HOIL1 showed significantly better disease-free survival (DFS) compared to those with lower levels of HOIP/HOIL1 (Supplementary Fig. 6S and S6T). Consistently, high RNF31/HOIP levels significantly correlated with prolonged DFS in TNBC patients who received systemic chemotherapy (Fig. 6j). Taken together, these data further support that increased LUBAC levels are strongly associated with better response to chemotherapy in TNBC patients, and OTULIN may promote drug resistance, at least in part, by diminishing LUBAC-mediated β-catenin linear ubiquitination and activating Wnt signaling in TNBC cells.

## Discussion

Wnt/β-catenin signaling is maintained at "Off" state by efficient negative regulatory mechanisms that limit the cytosolic pool of β-catenin through destruction complex-promoted proteasome-dependent degradation of β-catenin. The β-TrCP-SCF E3 ligase-mediated K48 ubiquitination of β-catenin is essential for keeping the Wnt/β-catenin activation in check[1]. Here we demonstrated that β-catenin is also attached with M1-linked polyubiquitin catalyzed by the LUBAC ligase complex, which enhances β-catenin degradation. This observation is consistent with a previous report showing that overexpressing LUBAC suppressed Wnt/β-catenin activation[11]. Moreover, our data indicated that linear ubiquitination promotes β-catenin association with β-TrCP in resting cells, thereby promoting its proteasomal degradation. Upon genotoxic treatment, DNA damage apical kinase DNA-PK could activate c-Abl which subsequently phosphorylates OTULIN at Tyr56. The Tyr56 phosphorylation promotes OTULIN dissociation from HOIP while enhancing its interaction with β-catenin. Increased OTULIN/β-catenin interaction leads to decreased linear ubiquitination of β-catenin and reduced β-catenin association with β-TrCP-SCF ligase, resulting in decreased K48 ubiquitination and stabilization of β-catenin. Surprisingly, we found Wnt receptor proteins LRP5/6 and adapters DVL1/2/3 are dispensable for DNA damage-induced β-catenin stabilization, suggesting that genotoxic Wnt/β-catenin

activation may be regulated in a Wnt ligand/receptor-independent manner. In line with this notion, inhibiting Porcupine, an essential palmitoyl transferase for Wnt protein lipidation and receptor binding, did not reduce Wnt/β-catenin activation by genotoxic agents. Taken together, our data support a critical role of OTULIN in promoting DNA damage-induced Wnt/β-catenin activation through disassembling M1 ubiquitin chains attached to β-catenin (Fig. 6j).

A previous report showed that OTULIN plays a critical role in mediating Wnt signaling during mouse embryo development[11]. Mutations of OTULIN/Gumby diminishing its catalytic activity were embryonic lethal and mice harboring these mutations showed abnormal facial nerve sprouting and embryonic angiogenic deficits such as disorganized branching vascular networks in the trunk and underdeveloped hierarchical vascular network in the head. Defective Wnt/β-catenin activation due to OTULIN mutation plays an important role in these development defects. Otulin was shown to interact with DVL2, whereas the biological significance of this interaction in regulating Wnt signaling was unclear. We found genotoxic agents activate Wnt/β-catenin in DVL1/2/3 triple knockout cells (Fig. 1f), indicating DVL2 is dispensable for DNA damage-induced Wnt activation. Moreover, ectopic OTULIN expression increased Wnt/β-catenin activity in the DVL-TKO cells (Fig. 2e), supporting a DVL2-independent role of OTULIN in promoting Wnt/β-catenin activation. Consistently, OTULIN overexpression also enhanced β-catenin stability and transactivity in LRP5/6-DKO cells (Fig. 2d), further supporting a Wnt canonical receptor-independent mechanism underlying OTULIN-promoted Wnt/β-catenin activation. However, there were additional transmembrane proteins, such as ROR, RYK, and GPR124, have been implicated in transducing Wnt signaling[1]. It is unclear whether these molecules are involved in DNA damage and/or OTULIN-promoted Wnt/β-catenin activation, which warrants further investigation.

Our data suggest that DNA-PK/ABL1-mediated OTULIN Tyr56 phosphorylation is not required for Wnt3a-induced Wnt/β-catenin activation. However, the decrease of β-catenin M1-ubiquitination was dramatically delayed in Wnt3a-treated cells compared to that by genotoxic treatments (Supplementary Fig. 4I). We speculate that the genotoxic stress-induced OTULIN phosphorylation at Tyr56 may not only increase the interaction between OTULIN and β-catenin but also enhance the LUBAC autoubiquitination and self-inhibition through dissociation from HOIP[43], which render OTULIN highly efficient in promoting β-catenin deubiquitination and stabilization upon DNA damage. It remains to be determined whether alternate kinase could be activated and/or phosphorylation-independent OTULIN activation may be induced downstream of Wnt receptor engagement, which may regulate OTULIN-mediated Wnt/β-catenin activation by Wnt ligands. We showed that K133 is essential for LUBAC-promoted linear ubiquitination of β-catenin. Intriguingly, β-catenin K133 was shown to be methylated by SMYD2 which is required for Wnt/β-catenin activation[44]. LUBAC-mediated linear ubiquitination on K133 may block the SMYD2-dependent β-catenin methylation. Besides stabilizing β-catenin, deubiquitinating β-catenin by OTULIN at Lys133 may also enable the subsequent methylation of Lys133 by SMYD2, which collaboratively promoted Wnt/β-catenin activation. It is also worth noting that embryos of OTULIN-C129A knock-in mice were resorbed at embryonic day (E)10.5, which is earlier than the time of embryonic death (E12.5–E14) observed in *Gumby* mice (W96R or D336E)[11,21]. Although all these OTULIN-inactive mutations caused vasculature abnormality in embryos, the C129A mice die of aberrant cell death rather than defective Wnt signaling. It is plausible that these inactivating mutations suppressed Otulin/Gumby activity to a different extent which dysregulated multiple

signaling cascades at different stages of embryo development. The hierarchy of these signaling processes during development and their reliance on Otulin activity remain to be delineated.

LUBAC-mediated linear ubiquitination plays critical roles in mediating immune response and cell death by modulating molecular signaling events such as NF-κB activation, ERK activation, inflammasome formation, RIG-I signaling, type I interferon response and necroptosis[16,21]. Although M1-linked polyubiquitination predominantly regulates signal transduction in physiological and pathological processes, linear ubiquitination by LUBAC has been associated with protein stability and proteasomal degradation. LUBAC was shown to preferentially bind to activated conventional PKCs and promote their linear ubiquitination, leading to their degradation and attenuation of PKC activity[45]. Linear polyubiquitination of misfolded Huntingtin protein facilitates its degradation by the proteasome[46]. We found OTULIN-dependent removal of the M1 chains on β-catenin also diminished its polyubiquitination with K48-linked chains (Fig. 3f). Accordingly, inhibiting LUBAC with gliotoxin reduced both M1- and K48-linked polyubiquitination of β-catenin. Meanwhile, LUBAC promoted β-catenin degradation was inhibited by a dominant-native mutant of β-TrCP (Fig. 3i), and blocking linear ubiquitination of β-catenin reduced its association with β-TrCP (Fig. 3j, k). Our results suggest that linear ubiquitination may indirectly promote proteasomal degradation of substrate proteins through enhancing their K48-linked polyubiquitination, which may be achieved by increasing the interaction between the substrate protein and the E3 ligase assembling K48-linked ubiquitin chains. Activated PKC was reported to be ubiquitinated with K48-linked chains by VHL (von Hippel-Lindau tumor-suppressor protein)-associated E3 ligase complex which directed PKC degradation[47]. Both CHIP (C terminus of Hsp70-interacting protein) and UBE3A (also known as E6AP) were shown to promote mutant Huntingtin protein degradation by facilitating its K48 ubiquitination[48,49]. It remains to be determined whether linear ubiquitination of activated PKC or misfolded Huntingtin also facilitates the K48-ubiquitination of the respective protein through these K48-specific E3 ligases.

DNA damage has been shown to induce nuclear accumulation and activation of c-Abl[33]. In the absence of DNA damage, c-Abl shuttles between the cytoplasm and nucleus[32]. Upon genotoxic treatment, cytoplasm-sequestered c-Abl is released from the binding of the 14-3-3 protein and translocates to the nucleus where it can be phosphorylated by DNA damage apical kinase ATM and DNA-PK[34,50]. Nuclear activated c-Abl is required for phosphorylation of histone acetyltransferase Tip60/KAT5, which could subsequently acetylate ATM and c-Abl, resulting in propagation of the DNA damage response signaling[32]. We found c-Abl is responsible for the Tyr56 phosphorylation of OTULIN upon DNA damage. The c-Abl-dependent OTULIN phosphorylation in response to genotoxic treatments relies on DNA-PK but not ATM or ATR. In TNBC cells treated with chemotherapeutic drugs, we detected nuclear localization of c-Abl whereas OTULIN nuclear translocation was not observed. Therefore, it is plausible that genotoxic stress-induced nuclear accumulation enables DNA-PK-dependent c-Abl activation, which may shuttle back to the cytoplasm where it phosphorylates OTULIN and promotes subsequent Wnt/β-catenin activation. Although ATM is required for DNA-PK-mediated c-Abl phosphorylation and activation by IR[51], inhibiting DNA-PK, but not ATM, abolished OTULIN Tyr56 phosphorylation in response to genotoxic drug treatment. Thus, DNA-PK-dependent c-Abl phosphorylation is essential for its activation and OTULIN phosphorylation at Tyr56 in response to genotoxic drugs, which plays a critical role in promoting Wnt/β-catenin activation and development of drug resistance in TNBC upon chemotherapy.

Breast cancer patients with basal-like subtype or triple-negative breast cancer generally have a worse prognosis than patients with other breast cancer subtypes due to innate and adaptive therapeutic resistance accompanied by aggressive metastasis[52]. Although TNBC patients overall are sensitive to cytotoxic chemotherapy initially, a significant number of TNBC patients rapidly develop drug resistance. TNBC was recently shown to adapt to neoadjuvant chemotherapy by adopting a reversible drug-tolerant state, in which a number of common gene signatures, such as glycolysis, MYC signaling, and EMT, were enriched[53]. We found that OTULIN is overexpressed in breast tumors, especially in the basal-like subtype (Supplementary Fig. 6L). Among the six molecular subtypes of TNBC[40], OTULIN levels are higher in BL2 and M subtypes which are enriched with increased Wnt/β-catenin gene signature (Supplementary Fig. 6M). Interestingly, GSEA analyses of genes, whose expression is significantly correlated with the OTULIN levels in breast cancer patients from TCGA-panCAN dataset, revealed that OTULIN expression is positively correlated with increased Wnt/β-catenin-target genes and MYC-target genes (Supplementary Fig. 6N–O), suggesting that OTULIN may play an important role in enhancing Wnt/β-catenin activation and MYC signaling. Additionally, increased OTULIN levels were observed in PDX samples from chemo-refractory TNBC patients compared to those who were responsive to chemotherapy. These data from breast cancer patients indicate that OTULIN-promoted Wnt/β-catenin activation may be required for cancer cells to adapt to the exposure of cytotoxic chemotherapeutics and develop drug resistance. There is substantial evidence that the outgrowth of metastatic lesions and CSCs is promoted by Wnt/β-catenin signaling. Wnt-regulated pathological processes, such as EMT and CSCs enrichment, have been associated with therapeutic resistance and metastasis[54–56]. We found that OTULIN overexpression significantly increased TNBC cell resistance to chemotherapy. Inhibiting Wnt/β-Catenin signaling or OTULIN sensitized TNBC xenografts to Doxorubicin treatment and reduced lung metastasis in vivo (Fig. 6a–e). Importantly, high OTULIN levels are associated with shorter OS and DFS in breast cancer patients (Supplementary Fig. 6). Altogether, our findings suggest that OTULIN may increase TNBC therapeutic resistance and aggressiveness by promoting Wnt/β-Catenin activation in response to chemotherapy. Targeting OTULIN and the chemotherapy-induced Wnt/β-Catenin activation may serve as a promising strategy to mitigate drug resistance and reduce metastasis in TNBC patients.

## Methods

**Animal studies**. Six-week-old immunocompromised NOD Scid Gamma (NSG) female mice were used and maintained under AAALAC-accredited specific pathogen-free housing vivarium and care, and veterinary supervision following standard guidelines for temperature and humidity with 12/12 light cycle. All procedures involving mice and experimental protocols were approved by the Institutional Animal Care and Use Committees (IACUC) of University of Tennessee Health Science Center. For xenograft models, 6-week immunocompromised NOD Scid Gamma (NSG) females were used. $10^6$ LM2-luc or LM2-luc-OTULIN-KO cells were suspended in 100 μl PBS for each mammary fat pad injection. When tumors grew close to 100 mm³, the mice were i.p. injected with Natural Saline (NS) as control, or Dox (1.5 mg/kg/Week), ICG-001 (100 mg/kg/2 days), or combination of Dox and ICG-001. Primary tumors were measured by caliper every other day until 4 weeks. Lung metastasis was monitored by bioluminescent imaging in the fourth week. At endpoints, tumors were dissected and weighed, and preserved for subsequent Western blot and qPCR examination.

**Cell culture**. Human breast cancer cell lines MDA-MB-231, MDA-MB-468, MCF7, and LM2, Human embryonic kidney 293 T and 293 A, Mouse Embryo Fibroblasts were grown in DMEM supplemented with 10% FBS and pen/strep. Human Breast cancer HCC1937 cells and HBRT1071 PDX cells were grown in RPMI-1640 supplemented with 10% FBS and pen/strep. Mouse FVB-MMTV-PyMT cells were grown in DMEM/F12 supplemented with 2% FBS and pen/strep. HCI PDX cells were grown in a special culture media composed of DMEM/F12 supplemented with 2% FBS, 10 mM HEPES, 1x Insulin-transferrin-selenium,

1x Antibiotic-Antimycotic, 5 ng/ml EGF, 0.3 μg/ml Hydrocortisone, 0.5 ng/ml Cholera toxin, 5 nM 3,3',5-triido-L-thyronine, 5 μM Isoproterenol hydrochloride, 50 nM Ethanolamine and 50 nM O-phosphorylethanolamine. Gene knockout cells LM2-luc_OTULIN-KO, MDA-MB-231_ABL1-KO, MDA-MB-231_OTULIN-KO, HEK293T_OTULIN-KO and 293T_HOIP-KO were generated with CRISPR-Cas9. FVB-MMTV-PyMT-Flag-OTULIN stable cell line was generated with lentiviral transduction.

**Viral transduction of cell lines and generation of stable cell line**. Lentiviral plasmids, Vsv-g, pMDL-g, and RSV-REV were transfected together into HEK293T cells to produce viruses. After 72 h, culture media was harvested and centrifuged at 1800 rpm, 5 min at room temperature to remove cell debris, and subsequently go through 0.45 μm filter. Then, the filtered viral media plus 6 μg/ml polybrene was used to infect target cells that were seeded one day before. The target cells were selected with puromycin (2 μg/ml) after 24 h transfection. Single colonies were picked after selection.

**qRT-PCR analysis**. Total RNA was isolated with TRIzol (Invitrogen) and retro-transcribed with a first-strand cDNA synthesis kit (Thermo Scientific). Real-time PCR analyses were performed using the PowerUp SYBR master mix (Applied Biosystems). The housekeeping gene GAPDH was used as an internal control.

**Immunoprecipitation and immunoblotting**. For IP and co-IP experiments, cells were collected in 1.5 ml EP tube from 10 cm dish after treated and lysed on ice for 30 min in IP lysis buffer (20 mM Tris (pH 7.0), 150 mM NaCl, 1 mM EDTA, 1 mM EGTA, 0.5% Nonidet P-40, 2 mM DTT, 0.5 mM PMSF, 20 mM β-glycerol phosphate, 1 mM sodium orthovanadate, 1 μg/ml leupeptin, 1 μg/ml aprotinin, 10 mM p-nitrophenyl phosphate, and 10 mM sodium fluoride). 5% supernatant lysate was taken as input and the rest was incubated with 2 μg of primary antibody and protein G-Sepharose at 4 °C overnight. Protein G-Sepharose-enriched complexes were resolved on SDS-PAGE gels and transferred onto PVDF membranes. The protein signals were visualized by ECL exposure.

To detect protein ubiquitination, cells were lysed with 1% SDS in IP lysis buffer at 95 °C for 30 min. The cell lysates were then diluted with IP lysis buffer to reduce SDS to 0.1% and mixed with primary antibodies and protein G-Sepharose for incubation at 4 °C overnight. The IP samples were washed five times with lysis buffer, separated by SDS-PAGE, transferred to nitrocellulose membrane, and boiled with distilled water for 30 min. The ubiquitin modification of precipitated proteins was examined by immunoblotting.

**Cell viability assay**. Cells were seeded in 96-well plates in triplicate at an optimal density. After indicated treatment, cell viability was evaluated by adding CCK8 reagents (Dojindo) with incubation for 2 h and plates were measured at 490 nm absorbance with a 96-well plate reader.

**c-Abl kinase assay**. A 10 cm dish MDA-MB-231 cells were treated as indicated. Pellets were lysed in IP lysis buffer. Supernatants were added with 2 μg of anti-c-Abl antibody for each tube. Samples were rotated for 1 h at 4 °C. Protein G-Sepharose beads were then added to each tube and the samples were rotated for 2 h at 4 °C. Subsequently, beads were collected and washed four times in lysis buffer and two times in c-Abl kinase buffer (25 mM Tris-HCL (pH 7.4), 10 mM MgCl2, 1 mM dithiothreitol). The c-Abl kinase activity was assayed in kinase buffer with 1 μg of substrate (GST-WT-PIM_1-80, GST-Y56F-PIM_1-80 or GST-Crkl) and 1 μCi of [γ-32P] ATP for 1 h at 30 °C. The reaction was terminated by adding 2x SDS loading buffer and boiled for 15 min. The proteins were separated by SDS-PAGE and then transferred to PVDF membranes which exposed to auto-radiography or Phosphoimage cassette, and the radioactive signal was analyzed by Cyclone PhosphoImager. The membranes were subjected to Western blotting with the indicated antibodies.

**Wnt signaling reporter assay**. Cells for Wnt signaling reporter assay were cotransfected with 10 ng of the pRLtkLuc plasmid and 0.2 μg of the Super8X TopFlash plasmid per well in triplicate in 24-well plate. The dual-luciferase assay was performed by using a Dual-Luciferase reporter assay system (catalog# E1960; Promega) according to the manufacturer's instructions. Luminescence was measured using a Promega GLOMAX 20/20 luminometer. Data were analyzed with Microsoft Excel.

**Tyrosine kinase siRNA library screen**. MDA-MB-231 cells were plated (5000 cells each well) in 96-well plate and reverse-transfected with Tyrosine kinase siR-NAs (Dharmacon, G-103100) distributed in each well. After 48 h, cells were treated with DMSO or Dox (2 μg/ml) for 90 min. Then, the medium was removed and cells were washed with PBS twice. 2x SDS was added to each well and lysate was transferred to a tube and heated at 95 °C for 10 min. Samples were resolved on SDS-PAGE gels and transferred onto PVDF membranes. P-OTULIN (Tyr56), OTULIN, and Tubulin signals were visualized by ECL exposure. The screen was repeated independently three times. Each blot was quantified with ImageJ and the

relative p-OTULIN level was calculated in mean ± SD with Excel. The result was presented in the volcano diagram.

**UbiCREST deubiquitinase enzyme assay**. MDA-MB-231 cells were treated with MG132 (10 μM) for 16 h and lysed in the same way as described in Immuno-precipitation. Supernatant from centrifuged lysate was incubated with 2 μg of anti-β-Catenin primary antibody and protein G-Sepharose at 4 °C overnight. The next day, G-Sepharose was washed five times in IP buffer and incubated with 1 U of indicated DUB in UbiCREST Kit (Boston Biochem)-provided reaction buffer at 37 °C. After 1 h, 2× SDS was added to terminate the reaction and samples were resolved by SDS-PAGE. The ubiquitin modification of precipitated proteins was examined by immunoblotting.

**In vitro ubiquitination assay**. Recombinant WT and K133R GST-β-Catenin (amino acid 1-153) were expressed in BL21 E. coli and purified using glutathione sepharose. Myc-HOIP was expressed in MDA-MB-231 cells and immunoprecipitated by using Myc-tag antibody and A/G agarose beads to generate active LUBAC complex. Briefly, reaction mixtures contained ubiquitin (250 μg/ml), E1 (5 μg/ml), E2 (10 μg/ml), precipitated LUBAC complex and the substrate GST-β-Catenin (100 μg/ml) and were incubated at 37 °C for 2 h in reaction buffer containing 20 mM Tris–HCl (pH 7.5), 5 mM $MgCl_2$, 2 mM DTT and 2 mM ATP. After the reaction, GST-β-Catenin fragment was pulldown by glutathione sepharose and subjected to SDS-PAGE and subsequent immunoblotting.

**Immunofluorescence**. MDA-MB-231 cells were seeded on glass coverslips at the optimal density. After Dox treatment, cells were fixed in Methanol at 4 °C for 20 min and then permeabilized for 5 min with 0.2% Triton X-100. Cells were blocked by 3% bovine serum albumin/PBS for 30 min before incubated with β-Catenin primary antibody (1: 200 in 1% BSA) overnight at 4 °C. The next day, cells were stained with DyLight 488 fluorescence-conjugated secondary antibody for 1 h. The nuclei were counterstained with 4′,6-diamidino-2-phenylindole (DAPI), and the cells were imaged with an EVOS immunofluorescence microscope (Life Technologies).

To observe the subcellular location of c-Abl and OTULIN after Dox treatment, cells were transfected with GFP-OTULIN for 24 h and exposed to Dox for the indicated time. Then, cells were fixed and blocked as above before incubated with c-Abl primary antibody (1:100 in 1% BSA) overnight at 4 °C. The second day, cells were stained with DyLight 594 and DAPI before imaging.

**Clonogenic assay**. MDA-MB-231 WT, OTULIN-KO, or OTULIN-KO cells reconstituted with OTULIN (WT or C129S mutant) or β-Catenin SA mutant were treated with 2 μg/ml Dox for 24 h. Then, cells were fixed with Methanol at −20 °C for 20 min and stained with 0.5% crystal violet in Methanol for 5 min. After washing off residue dye, the cell plate was imaged. Experiments were carried out at 3 times and Image J was used to quantify the stained area for each well. Cell survival percentage was calculated by the stained area in each treatment divided by that in control WT MDA-MB-231.

**Clinical data analysis**. The qPCR analysis of OTULIN was performed in breast cancer patient tumor samples and normal breast tissues at Fudan University Shanghai Cancer Center (FUSCC) and with tumor cDNA library from OriGene (Rockville, MD). Biopsy samples were collected under a published protocol[57], which was reviewed and approved by an independent ethical committee/institutional review board (IRB) at FUSCC, and all patients gave their written informed consent before inclusion in this study. Patient specimens to generate the PDXs 1071 and 1150 were collected under an IRB protocol approved by the UTHSC IRB. TNBC specimen 1071 was obtained from 57 years old women of African American descent and the TNBC specimen 1150 was collected from 74 years old women of European ancestry. Tumor specimens (1 $mm^3$) were implanted in female NSG mice and the growth was monitored. Once the tumors grew to 1000 $mm^3$, the tumors were either further passaged for experiments or were frozen in liquid nitrogen for further use. Once the PDX was generated, 1071 tumors were collected, and a cell line was created.

OTULIN transcription levels in samples from TCGA-BRCA genomic dataset were downloaded from Firehose Broad GDAC. Genes whose expression significantly correlated (Spearman's $R \geq 0.4$ or $\leq -0.4$, $p < 0.001$) with OTULIN levels in breast cancer patients within the TCGA-PanCAN study were retrieved through cBioPortal, enriched gene signatures were identified with GSEA. Overall survival and progression-free survival in combined breast cancer dataset (TCGA-TARGET-GTEx) were retrieved through UCSC-Xena, and compared between patients with high ($n = 2564$, top quantile) and low ($n = 2658$, bottom quantile) OTULIN expression. Disease-free survival in basal breast cancer patients (GSE21653) was analyzed between patients stratified by medium OTULIN level. Distant metastasis-free survival in breast cancer patients who received chemotherapies was analyzed through KM-plot[42].

**Quantification and statistical analysis**. Statistical parameters, including average, standard deviation, and statistical significance are presented in the figures.

Statistical analysis has been performed using Prism GraphPad 7 software. Two-sided Student's $t$-test was used to compare differences between two groups and the two-way ANOVA test was used to compare differences among multiple groups. $p < 0.05$ is considered significant, indicated as *$p < 0.05$, **$p < 0.01$, and ***$p < 0.001$.

**Reporting summary**. Further information on research design is available in the Nature Research Reporting Summary linked to this article.

## Data availability
The databases used in this study included Firehose Broad GDAC (https://gdac.broadinstitute.org/), cBioPortal (http://www.cbioportal.org/), Xena (https://xena.ucsc.edu/) and KM-plot (https://kmplot.com/analysis/). The source data underlying Figs. 1–6 and Supplementary Figs. 1–6 are provided as a Source Data file. All remaining relevant data are available in the article, Supplementary Information, or from the corresponding author upon reasonable request.

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

## Acknowledgements
We thank Drs. K.L. Guan, S. Angers, Z.J. Chen, J. Massagué, K. Iwai, J. Yue, and T.N. Seagroves for providing valuable reagents, and Drs. L. Pfeffer, R. Laribee for stimulating discussion and critical reading of the manuscript. Supports for this work are in part through Bridge fund, Memphis Institute of Regernerative Medicine, and CORNET award from UTHSC to Z.-H.W.

## Author contributions
W.W. and Z.-H.W. designed and conceptualized the research, W.W., M.L., Y.C., J.X., and B.F. conducted most of the experiments and analysis of the data.; W.W., M.F., G.M.-C., R.N., J.W., and Z.-H.W. analyzed and interpreted the data, W.W., M.F., G.M.-C., R.N., J.W., and Z.-H.W. wrote and edited the paper. All authors discussed the results and reviewed the manuscript.

## Competing interests
The authors declare no competing interests.
