## [Peer Review File · Nature Communications]

Reviewers' Comments:

Reviewer #1:

Remarks to the Author:

In the manuscript submitted by Wang et al., the authors provide evidence for the DNA damage-induced activation of Wnt/ β -catenin signaling through a mechanism which is independent of canonical signaling through FZD/LRP heterodimeric receptors and Wnt ligands. They show that genotoxic stress induces the phosphorylation of OTULIN at Tyr56, which in turn allows for its engagement with β -catenin. This inhibits linear ubiquitination of β -catenin by LUBAC and subsequent Lys48-linked polyubiquitination by β -TRCP, thereby attenuating the proteasomal degradation of β -catenin upon DNA damage. Furthermore, through a limited siRNA screen with a tyrosine kinase sub-library, they identified ABL1 as the kinase responsible for phosphorylating OTULIN upon genotoxic stress. OTULIN-mediated Wnt/ β -catenin activation upon genotoxic stress promotes drug resistance in TNBC tumors, thus suggesting that targeting OTULIN may be an appropriate therapeutic strategy to reduce metastasis and drug resistance in TNBC patients. While most experiments are thoroughly executed, the following additional experiments should be considered to warrant publication:

Major points:

1. The finding that Dox treatment induces the association between endogenous OTULIN and β -catenin is very subtle (i.e. Fig 4A), although this is a major part of the mechanism for OTULIN-mediated activation of Wnt signaling. It would be helpful to have a single figure comparing relative binding of β -catenin to OTULIN, HOIP, and β -TRCP with and without Dox treatment to better appreciate how these differences in binding contribute to the mechanism for β -catenin stabilization. For example, in the model in Figure 6J, the authors suggest that LUBAC is bound to β -catenin regardless of whether Wnt signaling is turned on and OTULIN is bound. Would increasing the concentration of LUBAC in a dose-dependent manner outcompete OTULIN for access to β -catenin for linear ubiquitination? Also, the model indicates that β -TRCP is bound to β -catenin in both the ON and OFF states, yet the data in Figure 3 showed that β -TrCP binding was reduced in response to genotoxic stress. The model should be re-drawn to better emphasize this point.
2. The model in Figure 6J also indicates that c-Abl is translocated from the nucleus to phosphorylate OTULIN in the cytoplasm, although no direct evidence was provided for this. The authors should include immunofluorescence data showing the nuclear localization of c-Abl and OTULIN with or without Dox treatment to better clarify this point.
3. In addition to c-Abl, ITK was also found in the tyrosine kinase screen (Fig. 5A) to induce the phosphorylation of OTULIN, yet it was not followed up on. The authors should provide a supplemental figure to demonstrate that ITK phosphorylates OTULIN but does not impact Wnt-signaling in combination with c-Abl, as they suggest.
4. In Figure 6, the authors show a correlation between OTULIN levels and drug resistance in TNBC cells. It would also be interesting to observe whether there is also any correlation with LUBAC expression levels in the TCGA dataset. Does modulation of LUBAC attenuate the effects of OTULIN on drug resistance in the TNBC cells?

Minor points:

1. Figure S3E has a misplaced K48-Ub data label.
2. The following sentence should reference Figure S5B instead of S5C: "In contrast, inhibiting c-Abl did not affect Wnt3a-induced β -catenin stabilization." (pg. 13)
3. Methods section is not complete, no info on where the pY56-Otulin antibody came from and no info on sources of chemicals and antibodies, cell lines, etc

Reviewer #2:

Remarks to the Author:

This manuscript describes the underlying mechanism that Wnt/ β -catenin signaling is activated by genotoxic agents in breast cancer cells. The authors showed that OTULIN, a linear linkage-specific deubiquitinase, is required for the DNA damage-induced β -catenin activation. OTULIN was phosphorylated at Tyr56 by ABL1 and activated, resulting in mediating deubiquitination of the M1 ubiquitin chain of β -catenin at lysine 133. M1-linked linear ubiquitination of β -catenin by LUBAC promoted the interaction of β -TrCP with β -catenin, thereby inducing K48-linked polyubiquitination and degradation of β -catenin. In addition, the authors showed that OTULIN depletion decreased TNBC cells resistance to chemotherapeutics in vivo.

From these results, the authors proposed that OTULIN activates Wnt/ β -catenin signaling by inhibiting linear ubiquitination, K48-linked polyubiquitination, and proteasomal degradation of β -catenin, upon DNA damage. Although this paper is potentially interesting, there are several issues that the authors need to address.

Major comments;

(1) Is OTULIN-mediated deubiquitination of β -catenin involved in Wnt ligand-dependent activation of β -catenin signaling? Wnt3a did not induce OTULIN phosphorylation (sFig. 4G) and ABL1 or DNA-PK inhibitor that attenuates OTULIN phosphorylation and functions did not affect β -catenin stabilization by Wnt3a (sFig. 5B, J). In contrast, OTULIN KO inhibited Wnt3a-induced stabilization of β -catenin and target gene expression (Fig. 2A-C). These results are controversial, therefore, the authors should interpret carefully them.

(2) CYLD, a USP family deubiquitinase, binds the PUB domain of HOIP and cleaves M1- and K63-ubiquitin linkage. It is crucial to test whether CYLD regulates β -catenin levels in the presence of genotoxic agents and Wnt3a.

(3) In Fig. 3B, polyubiquitination of β -catenin seems to be greatly reduced by OTULIN or OTUB1, whereas K48-linked and M1-linked polyubiquitin chains, respectively, should not be affected by these deubiquitinases. The authors need to confirm by immunoblotting using chain type specific antibodies.

(4) In Fig. 3E, the authors should show the effect of genotoxic agents and Wnt3a on the enhanced M1-ubiquitination of β -catenin in OTULIN-KO cells. The authors suggested that linear ubiquitination of β -catenin may occur prior to K48 ubiquitination. Then, it is necessary to examine K48-linked ubiquitination and proteasomal degradation of β -catenin in HOIP-KO cells.

(5) The authors identified that K133 is a linear ubiquitination site in β -catenin. This reviewer is wondering what is the physiological roles of M1-linked linear ubiquitination of β -catenin at steady state? It has been reported that β -catenin Lys133 is methylated by SMYD2 and this modification is critical for the binding to FOXM1, nuclear translocation, and activation of Wnt signaling [PMID:28915556]. The authors need to test the relationship or interdependence between M1-linked linear ubiquitination and methylation at Lys133

(6) In sFig. 3E, some mutants of β -catenin other than K133R reduced M1-ubiquitination. The authors should examine the ubiquitination sites of β -catenin by mass spectrometry. Otherwise, in vitro ubiquitination assay is necessary to confirm that K113 is the M1-ubiquitination site.

(7) In related to Fig. 3K, the authors should show the stability of β -catenin K133R that has defect of binding to β -TrCP compared with β -catenin K19/49R and β -catenin SA mutant (S33A/S37A/T41A/S45A, which is known to be constitutively activated). In addition, does OTULIN affect β -catenin stabilization by GSK3 inhibitors such as CHIR99021 that inhibits β -catenin phosphorylation?

(8) In Fig. 3A and I, the authors showed that LUBAC expression induced M1-linked linear ubiquitination and degradation of β -catenin. To clarify that K19/K49 are K48-ubiquitination sites of β -catenin, whether LUBAC degrades β -catenin K19/K49R mutant should be tested.

(9) As the molecular anatomy, the ABL1-binding region of OTULIN should be determined. Furthermore, does the region overlap with β -catenin-binding region of OUTLIN?

(10) The authors should clarify the relationship between phosphorylation and M1-linked linear ubiquitination of β -catenin. In Fig. 3K, M1-ubiquitination defect (K133R) did not affect β -catenin phosphorylation at T41/S45 which is required for β -TrCP binding. Does the state of β -catenin phosphorylation at S33/S37 affect M1-ubiquitination of β -catenin? M1-ubiquitination state of β -catenin SA mutant (S33A/S37A/T41A/S45A) with or without DOX should be analyzed.

(11) The authors identified that ABL1 is responsible for phosphorylation of OTULIN, which regulates the Wnt/ β -catenin pathway in Fig. 5. However, many studies reported that OUTLIN is involved in NF- κ B signaling and cell death. Therefore, the authors need to test whether the NF- κ B, MAPK, and cell death pathways are activated by genotoxic agents and inflammatory cytokines in ABL1-KO cells.

(12) In Fig. 5E, ABL1 KO phenotypes should be rescued by expression of WT ABL1 but not ABL1 kinase dead mutant.

(13) In Fig. 5I, the total β -catenin level was unchanged by DOX treatment. The result is not consistent with other results such as Fig. 1A.

(14) In Fig. 6, direct evidence that in vivo phenotypes induced by OTULIN KO cells are mediated by Wnt/ β -catenin signaling was not shown. The authors should show that phenotypes induced by OTULIN KO are rescued by Wnt signaling activation such as the expression of β -catenin SA mutant.

Reviewer #1:

Major points:

1. The finding that Dox treatment induces the association between endogenous OTULIN and β -catenin is very subtle (i.e. Fig 4A), although this is a major part of the mechanism for OTULIN-mediated activation of Wnt signaling. It would be helpful to have a single figure comparing relative binding of β -catenin to OTULIN, HOIP, and β -TRCP with and without Dox treatment to better appreciate how these differences in binding contribute to the mechanism for β -catenin stabilization. For example, in the model in Figure 6J, the authors suggest that LUBAC is bound to β -catenin regardless of whether Wnt signaling is turned on and OTULIN is bound. Would increasing the concentration of LUBAC in a dose-dependent manner outcompete OTULIN for access to β -catenin for linear ubiquitination? Also, the model indicates that β -TRCP is bound to β -catenin in both the ON and OFF states, yet the data in Figure 3 showed that β -TrCP binding was reduced in response to genotoxic stress. The model should be re-drawn to better emphasize this point.

Following the reviewer's suggestion, we now include data showing the change of proteins binding to β -catenin in MDA-MB-231 cells upon Dox treatment as revised Fig 4B. We found substantially increased OTULIN association with β -catenin in Dox-treated MDA-MB-231 cells, whereas the interaction between β -catenin and β -TrCP was significantly decreased. We did not observe a remarkable change of the β -catenin-binding with the LUBAC complex in response to Dox treatment. We also show that increased LUBAC catalytic subunit HOIP enhanced the interaction between β -catenin and HOIP while minimally affecting β -catenin association with OTULIN (revised Figure S4C), supporting that HOIP and OTULIN bind to β -catenin independently. In response to the reviewer's suggestion, we have revised the model in Fig 6K to reflect the decreased interaction between β -catenin and β -TrCP in response to genotoxic stress.

2. The model in Figure 6J also indicates that c-Abl is translocated from the nucleus to phosphorylate OTULIN in the cytoplasm, although no direct evidence was provided for this. The authors should include immunofluorescence data showing the nuclear localization of c-Abl and OTULIN with or without Dox treatment to better clarify this point.

We carried out the immunofluorescence staining of ABL1 and OTULIN in MDA-MB-231 cells with or without Dox treatment. As shown in revised Fig S5H, in response to Dox treatment, significant amount of c-Abl translocated from the nucleus to cytoplasm where it colocalized with OTULIN. No substantial change of OTULIN cytoplasmic localization was observed in MDA-MB-231 cells treated with Dox. We also analyzed the subcellular localization of OTULIN and ABL1 following Dox treatment with fractionation analyses as in revised Fig S5I. Consistently, we observed increased ABL1 cytoplasmic distribution following the Dox treatment, which correlated with increased OTULIN Tyr56 phosphorylation in the cytoplasm. We did not observe the nuclear translocation of OTULIN in response to Dox treatment, suggesting the OTULIN was phosphorylated by activated ABL1 in the cytoplasm upon genotoxic stimulation.

3. In addition to c-Abl, ITK was also found in the tyrosine kinase screen (Fig. 5A) to induce the phosphorylation of OTULIN, yet it was not followed up on. The authors should provide a supplemental figure to demonstrate that ITK phosphorylates OTULIN but does not impact Wnt-signaling in combination with c-Abl, as they suggest.

Following the reviewer's suggestion, we have included data regarding ITK in revised Fig S5C. We showed that inhibiting ITK with BMS-509744 reduced OTULIN phosphorylation in response to Dox treatment, especially at later timepoint. However, Dox-induced β -catenin stabilization was not affected by ITK inhibition, which may be due to its delayed kinetics in inhibiting OTULIN phosphorylation. In contrast, inhibiting ABL1 with imatinib blocked both OTULIN phosphorylation and β -catenin accumulation by Dox treatment.

4. In Figure 6, the authors show a correlation between OTULIN levels and drug resistance in TNBC cells. It would also be interesting to observe whether there is also any correlation with LUBAC expression levels in the TCGA dataset. Does modulation of LUBAC attenuate the effects of OTULIN on drug resistance in the TNBC cells?

Thanks for the reviewer's suggestion. We analyzed the survival data of breast cancer patients following the systemic treatment (KM-plotter). As shown in revised Fig S6S and S6T, breast cancer patients with high levels of LUBAC subunit HOIP or HOIL1 showed significantly better DFS compared to those with lower levels of HOIP/HOIL1. Consistently, high levels of RNF31/HOIP (the catalytic subunit of LUBAC) significantly correlated with prolonged DFS in TNBC patients who received systemic chemotherapy (Fig 6J). These data further support that increased LUBAC levels are strongly associated with better response to chemotherapy in TNBC patients, and OTULIN may promote drug resistance by diminishing LUBAC-mediated linear ubiquitination in TNBC cells. As expected, we found overexpression of HOIP enhanced drug sensitivity in MDA-MB-231 cells. Moreover, HOIP overexpression also attenuated drug resistance induced by increased OTULIN levels (Fig S6R).

Minor points:

1. Figure S3E has a misplaced K48-Ub data label.

We have corrected the figure labels accordingly.

2. The following sentence should reference Figure S5B instead of S5C: "In contrast, inhibiting c-Abl did not affect Wnt3a-induced β -catenin stabilization." (pg. 13)

We have revised the text with the correct figure panel citation.

3. Methods section is not complete, no info on where the pY56-Otulin antibody came from and no info on sources of chemicals and antibodies, cell lines, etc

We have included all the reagent detail information in the revised method section and supplementary table 1.

Reviewer #2:

Major comments;

(1) Is OTULIN-mediated deubiquitination of β -catenin involved in Wnt ligand-dependent activation of β -catenin signaling? Wnt3a did not induce OTULIN phosphorylation (sFig. 4G) and ABL1 or DNA-PK inhibitor that attenuates OTULIN phosphorylation and functions did not affect

β-catenin stabilization by Wnt3a (sFig. 5B, J). In contrast, OTULIN KO inhibited Wnt3a-induced stabilization of β-catenin and target gene expression (Fig. 2A-C). These results are controversial, therefore, the authors should interpret carefully them.

We agree that OTULIN plays an important role in mediating canonical Wnt signaling activated by Wnt3a which likely does not require ABL1-dependent OTULIN phosphorylation. The Wnt3a-induced Wnt/β-catenin activation was reduced to a level of around 50% of that in WT MDA-MB-231 cells, although the induction was still significant (Fig. 2B). Meanwhile, genotoxic treatment-induced Wnt/β-catenin activation was completely abolished. The reduction of Wnt/β-catenin activation by canonical Wnt ligands in OTULIN-deficient cells could cause critical biological dysfunctions, which may contribute to the developmental defects observed in OTULIN/Gumby-deficient mice. Whether OTULIN regulates canonical Wnt signaling by modulating β-catenin linear ubiquitination or through DVL2 as suggested in the previous report (Rivkin E, 2013) warrants further investigation. Our data suggest that DNA-PK/ABL1-mediated OTULIN Tyr56 phosphorylation may be dispensable for Wnt3a-induced Wnt/β-catenin activation. It is possible that alternate kinase may be activated and/or phosphorylation-independent regulation of OTULIN activity may be induced downstream of Wnt receptor engagement, which regulates OTULIN-mediated Wnt/β-catenin activation by Wnt ligands. We have discussed the potential role and regulation of OTULIN in Wnt ligand-dependent Wnt/β-catenin in the discussion section (p20).

(2) CYLD, a USP family deubiquitinase, binds the PUB domain of HOIP and cleaves M1- and K63-ubiquitin linkage. It is crucial to test whether CYLD regulates β-catenin levels in the presence of genotoxic agents and Wnt3a.

We examined the influence of CYLD in regulating Wnt/β-catenin activation in both human and mouse cells. As shown in revised Fig. S2O-P, knockdown CYLD in MDA-MB-231 cells did not affect β-catenin stabilization by Dox or Wnt3a treatment. Accordingly, CYLD deficiency did not impair Dox-induced Wnt/β-catenin activation in MEFs. These data indicate that CYLD may be dispensable for Wnt/β-catenin activation by genotoxic stress and Wnt3a.

(3) In Fig. 3B, polyubiquitination of β-catenin seems to be greatly reduced by OTULIN or OTUB1, whereas K48-linked and M1-linked polyubiquitin chains, respectively, should not be affected by these deubiquitinases. The authors need to confirm by immunoblotting using chain type specific antibodies.

Following the reviewer's suggestion, we confirmed the linkage specificity of the β-catenin polyubiquitin chains and the DUB activity of OTULIN and OTUB1. As shown in revised Fig S3A, accumulated polyubiquitin chains attached to β-catenin by inhibiting proteasome were sensitive to OTULIN or OTUB1 treatment which removes M1- or K48-linked chains respectively.

(4) In Fig. 3E, the authors should show the effect of genotoxic agents and Wnt3a on the enhanced M1-ubiquitination of β-catenin in OTULIN-KO cells. The authors suggested that linear ubiquitination of β-catenin may occur prior to K48 ubiquitination. Then, it is necessary to examine K48-linked ubiquitination and proteasomal degradation of β-catenin in HOIP-KO cells.

In revised Fig 3E, we now show that Dox failed to decrease β-catenin linear ubiquitination in

OTULIN-KO MDA-MB-231 cells, which is increased compared to that in WT cells, supporting an essential role of OTULIN to catalyze β -catenin M1 ubiquitination. We also showed that the K48-ubiquitination of β -catenin was decreased along with M1-ubiquitination in HOIP-KO cells in Fig S3B, supporting an important role of linear ubiquitination in facilitating K48 ubiquitination of β -catenin.

(5) The authors identified that K133 is a linear ubiquitination site in β -catenin. This reviewer is wondering what is the physiological roles of M1-linked linear ubiquitination of β -catenin at steady state? It has been reported that β -catenin Lys133 is methylated by SMYD2 and this modification is critical for the binding to FOXM1, nuclear translocation, and activation of Wnt signaling [PMID:28915556]. The authors need to test the relationship or interdependence between M1-linked linear ubiquitination and methylation at Lys133

We appreciate that the reviewer raised a very interesting and important comment regarding K133 modification in Wnt signaling. In the previous report, the β -catenin Lys133 methylation was detected in cells overexpressing SMYD2 with a customized β -catenin K133me-specific antibody. We contacted the group but was not able to obtain the β -catenin K133me antibody. We then took an alternative approach by using an antibody recognizing mono-methylated lysine (Cell Signaling Tech, #14679S) to probe immunoprecipitated β -catenin (WT or K133R mutant) from cells exposed to Dox. As shown below, we were not able to detect convincing signaling of β -catenin methylation at Lys133 in this experimental setting. It is possible that the available antibody to us was not optimal for detecting β -catenin methylation. Alternatively, the methylation of Lys133 is primarily involved in SMYD2-dependent Wnt/ β -catenin activation. Nevertheless, it is plausible that OTULIN-mediated deubiquitination of β -catenin at Lys133 enables the subsequent methylation of Lys133 by SMYD2, which collaboratively promoted Wnt/ β -catenin activation. We have included the discussion of this potentially sequential modification of Lys133 in mediating Wnt/ β -catenin signaling in the revised manuscript (p20).

(6) In sFig. 3E, some mutants of β -catenin other than K133R reduced M1-ubiquitination. The authors should examine the ubiquitination sites of β -catenin by mass spectrometry. Otherwise, in vitro ubiquitination assay is necessary to confirm that K113 is the M1-ubiquitination site.

Ubiquitination on Lys133 of β -catenin has been reported in multiple previous mass spectrometry analyses (such as PMID: 21906893, 22790023, 23266961. Phosphosite.org). Therefore, we performed an in vitro ubiquitination assay using recombinant β -catenin₁₋₁₅₃ (WT and K133R mutant) as the substrate. As shown in revised Fig S3H, wildtype GST- β -catenin, but not K133R

mutant, was ubiquitinated by LUBAC complex precipitated from cells expressing wildtype HOIP, but not catalytic inactive HOIP-CS mutant. These data further confirmed that K133 is a critical residue for β -catenin linear ubiquitination by LUBAC.

(7) In related to Fig. 3K, the authors should show the stability of β -catenin K133R that has defect of binding to β -TrCP compared with β -catenin K19/49R and β -catenin SA mutant (S33A/S37A/T41A/S45A, which is known to be constitutively activated). In addition, does OTULIN affect β -catenin stabilization by GSK3 inhibitors such as CHIR99021 that inhibits β -catenin phosphorylation?

We observed CHIR-induced β -catenin stabilization in OTULIN-deficient cells, but at a substantially decreased level when compared to that in WT cells (revised Fig S3L). This data suggest that both GSK3-dependent phosphorylation and LUBAC-mediated linear ubiquitination contribute to the effective binding of β -catenin with β -TrCP and its subsequent degradation. Accordingly, K133R mutation remarkably enhanced the stability of β -catenin, which was comparable to the K19/49R and SA (S33A/S37A/T41A/S45A) mutation (Fig S3M).

(8) In Fig. 3A and I, the authors showed that LUBAC expression induced M1-linked linear ubiquitination and degradation of β -catenin. To clarify that K19/K49 are K48-ubiquitination sites of β -catenin, whether LUBAC degrades β -catenin K19/K49R mutant should be tested.

We have determined if LUBAC overexpression could promote the degradation of β -catenin K19/49R mutant. As shown in revised Fig S3J, K19/49R mutant was resistant to LUBAC-enhanced β -catenin degradation, further supporting a critical role of K48 ubiquitination in mediating β -catenin proteasomal degradation downstream of LUBAC-promoted M1 ubiquitination.

(9) As the molecular anatomy, the ABL1-binding region of OTULIN should be determined. Furthermore, does the region overlap with β -catenin-binding region of OUTLIN?

We mapped the OTULIN domain required for interaction with ABL1 as the reviewer suggested. As shown in revised Fig S5J, the C-terminal OTU domain is required for OTULIN interaction with ABL1, which is not overlapped with the N-terminal domain required for OTULIN interaction with β -catenin.

(10) The authors should clarify the relationship between phosphorylation and M1-linked linear ubiquitination of β -catenin. In Fig. 3K, M1-ubiquitination defect (K133R) did not affect β -catenin phosphorylation at T41/S45 which is required for β -TrCP binding. Does the state of β -catenin phosphorylation at S33/S37 affect M1-ubiquitination of β -catenin? M1-ubiquitination state of β -catenin SA mutant (S33A/S37A/T41A/S45A) with or without DOX should be analyzed.

As shown in revised Fig S3K, we found the degron phosphorylation of β -catenin was dispensable for its linear ubiquitination and Dox treatment effectively decreased linear ubiquitination of the β -catenin degron SA mutant (S33A/S37A/T41A/S45A), supporting a parallel role of M1-ubiquitination along with degron phosphorylation in promoting β -TrCP-dependent β -catenin ubiquitination. As we showed in OTULIN-KO cell, inhibiting GSK3 with CHIR99021 was able to stabilize β -catenin at a much reduced level (Fig S3L), all these data support that the

mutually independent but collaborative roles of LUBAC-promoted linear ubiquitination and GSK3-dependent phosphorylation of β -catenin in controlling its stability.

(11) The authors identified that ABL1 is responsible for phosphorylation of OTULIN, which regulates the Wnt/ β -catenin pathway in Fig. 5. However, many studies reported that OUTLIN is involved in NF- κ B signaling and cell death. Therefore, the authors need to test whether the NF- κ B, MAPK, and cell death pathways are activated by genotoxic agents and inflammatory cytokines in ABL1-KO cells.

Following the reviewer's suggestion, we analyzed the impact of ABL1 deficiency on TNF α or Dox-induced NF- κ B, MAPK, and cell death pathways as shown in revised Fig S5P-Q. ABL1 deletion did not affect TNF α -induced activation of NF- κ B, MAPK and apoptosis signaling, whereas Dox-induced NF- κ B activation was attenuated in ABL1-KO cells, which correlated with increased Caspase 3 cleavage in these cells upon Dox treatment. It is plausible that DNA damage-induced ABL1 activation also promotes genotoxic NF- κ B activation and suppresses apoptosis upon genotoxic stress. We will determine the underlying mechanism and the involvement of OTULIN phosphorylation in separate studies.

(12) In Fig. 5E, ABL1 KO phenotypes should be rescued by expression of WT ABL1 but not ABL1 kinase dead mutant.

In revised Fig 5E, we now show that the reconstitution of ABL1-WT but not a kinase-dead mutant (K290R) in ABL1-KO cells rescued the Dox-induced OTULIN phosphorylation, supporting an essential role of ABL1 in phosphorylating OTULIN upon DNA damage.

(13) In Fig. 5I, the total β -catenin level was unchanged by DOX treatment. The result is not consistent with other results such as Fig. 1A.

In Fig. 5I, we treated cells with MG132 to prevent β -catenin degradation which enabled us to observe changes of β -catenin interaction with OTULIN without being affected by uneven β -catenin levels in the different treatment condition. We have revised the figure legend to reflect this experimental detail.

(14) In Fig. 6, direct evidence that in vivo phenotypes induced by OTULIN KO cells are mediated by Wnt/ β -catenin signaling was not shown. The authors should show that phenotypes induced by OTULIN KO are rescued by Wnt signaling activation such as the expression of β -catenin SA mutant.

To validate the essential role of Wnt/ β -catenin in mediating OTULIN-promoted drug resistance in TNBC cells, we overexpressed constitutively active β -catenin SA mutant in OTULIN-KO cells. In a clonogenic assay of the MDA-MB-231 cells exposed to Dox treatment (Figure S6G), we found ectopic expression of β -catenin SA mutant abrogated the enhanced drug sensitivity in OTULIN-KO MDA-MB-231 cells and significantly enhanced TNBC cell survival after Dox treatment, which is comparable to the cells reconstituted with OTULIN-WT. These data indicated that decreased Wnt/ β -catenin activation by genotoxic drugs in OTULIN-deficient cells is essential for enhanced chemosensitivity, and forced β -catenin activation is sufficient to enhance the drug resistance in OTULIN-deficient cells.

Reviewers' Comments:

Reviewer #1:

Remarks to the Author:

The authors have addressed most of the major concerns raised by the reviewers. The paper is now appropriate for publication.

Reviewer #2:

Remarks to the Author:

According to our criticisms, the authors dramatically improved the manuscript. The findings are novel and the experiments are carefully done. Although I am almost satisfied with the authors' responses, I would like to ask one question again. As already commented in (1) and (4), the authors should show direct results indicating whether or not the state of M1-ubiquitination of β -catenin is affected by Wnt3a stimulation. The authors suggested that LUBAC-promoted linear ubiquitination has independent but collaborative roles of GSK3-dependent phosphorylation of β -catenin in controlling its stability. If Wnt stimulation directly affects linear ubiquitination of β -catenin, it would give a strong impact on the Wnt research field.

Reviewer #2 (Remarks to the Author):

According to our criticisms, the authors dramatically improved the manuscript. The findings are novel and the experiments are carefully done. Although I am almost satisfied with the authors' responses, I would like to ask one question again. As already commented in (1) and (4), the authors should show direct results indicating whether or not the state of M1-ubiquitination of β -catenin is affected by Wnt3a stimulation. The authors suggested that LUBAC-promoted linear ubiquitination has independent but collaborative roles of GSK3-dependent phosphorylation of β -catenin in controlling its stability. If Wnt stimulation directly affects linear ubiquitination of β -catenin, it would give a strong impact on the Wnt research field.

We agree with the reviewer that it is critical to determine if the status of β -catenin linear ubiquitination is affected by Wnt3a stimulation. In our revised Fig 3E, we noticed that the decrease of β -catenin M1-ubiquitination by Wnt3a treatment was less prominent compared to Dox treatment, which may be due to the increased total β -catenin level in the Wnt3a-treated sample compared to the control. Moreover, the β -catenin M1-ubiquitination level was increased in OTULIN-KO cells. We thereby carried out temporal analyses of β -catenin M1-ubiquitination after Wnt3a treatment with or without proteasome inhibitor MG132 pretreatment. As shown below, Wnt3a treatment decreases β -catenin M1-ubiquitination with slower kinetics, compared to that by Dox treatment (Fig 4H), which is more prominent in cells pretreated with MG132 (also as revised Fig S4I). These data suggest that Wnt3a stimulation decreases the level of β -catenin M1-ubiquitination with slower kinetics, which may also contribute to decreased β -catenin association with β -TrCP, resulting in reduced β -catenin K48-ubiquitination and its stabilization. The slower kinetics in Wnt3a-treated cells may be due to the absence of ABL1-dependent OTULIN phosphorylation as in cells treated with genotoxic agents. Nevertheless, these data support that the OTULIN-mediated decrease of β -catenin M1-ubiquitination also plays an important role in Wnt3a-induced β -catenin stabilization and activation, which is consistent with the substantially decreased Wnt3a-induced β -catenin transactivation in OTULIN-depleted cells (Fig 2B, 2C). We have included these observations and discussion on Pg. 12 and 20, and will further explore the regulatory mechanisms in our future studies.

Reviewers' Comments:

Reviewer #2:

Remarks to the Author:

The new data supports the important finding that Wnt reduces linear ubiquitination of β -catenin, thereby controlling Wnt signaling. This reviewer has no further question.